# Navigating the Latent Space Dynamics of Neural Models

**Marco Fumero**[*]
IST Austria

**Luca Moschella**[†]
Sapienza

**Emanuele Rodolà**[‡]
Sapienza / Paradigma

**Francesco Locatello**[‡]
IST Austria

## Abstract

Neural networks transform high-dimensional data into compact, structured representations, often modeled as elements of a lower dimensional latent space. In this paper, we present an alternative interpretation of neural models as dynamical systems acting on the latent manifold. Specifically, we show that autoencoder models implicitly define a *latent vector field* on the manifold, derived by iteratively applying the encoding-decoding map, without any additional training. We observe that standard training procedures introduce inductive biases that lead to the emergence of attractor points within this vector field. Drawing on this insight, we propose to leverage the vector field as a *representation* for the network, providing a novel tool to analyze the properties of the model and the data. This representation enables to: $(i)$ analyze the generalization and memorization regimes of neural models, even throughout training; $(ii)$ extract prior knowledge encoded in the network's parameters from the attractors, without requiring any input data; $(iii)$ identify out-of-distribution samples from their trajectories in the vector field. We further validate our approach on vision foundation models, showcasing the applicability and effectiveness of our method in real-world scenarios.

## 1 Introduction

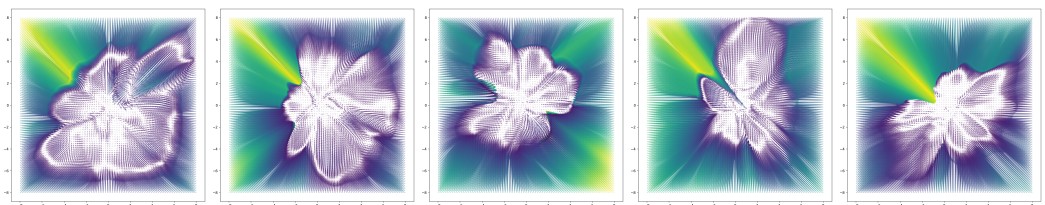

Figure 1: *Latent dynamics of AEs.* Latent vector fields induced by autoencoders with bottleneck $k = 2$, trained on MNIST, with $\mathbf{z}_0 \sim \mathcal{U}[-8, 8]$. Models with different initializations are shown. Colors (viridis colormap) represent vector norms ranging from violet (low) to yellow (high). The shape of the latent manifold identifies with the encoder's support. White regions indicate where the vector field vanishes, revealing attractors aligned with high-density areas of the data distribution.

Neural networks are powerful function approximators, capable of solving complex tasks across a wide range of domains (Jumper et al., 2021; Radford et al., 2021). A core component of their success lies in their ability to transform high-dimensional inputs into compact, structured representations (Bengio et al., 2013), typically modeled as points in a lower-dimensional latent space.

In this work, we propose a novel perspective: for any given autoencoder (AE) architecture, there exists an associated *latent vector field* (see Figure 1) that characterizes the model's behavior in

---

[*]Corresponding email: marco.fumero@ist.ac.at
[†]Currently at Apple.
[‡]Equally advising

representation space. This field arises naturally by iterating the encoder-decoder map in the latent space, requiring no additional training. Intuitively, such dynamics often settle into *attractors*: stable states toward which nearby trajectories converge, summarizing the long-term behavior of the system.

While prior work has shown that overparameterized autoencoders can memorize training data under strong assumptions (Radhakrishnan et al., 2020; Jiang and Pehlevan, 2020), we consider a broader setting. We show that memorization emerges as just *one* type of attractor in the latent vector field, and that the structure of this field reflects deeper properties of the network, such as generalization behavior and sensitivity to distribution shifts.

Our contributions can be summarized as follows:

- We show that every AE implicitly defines a latent vector field, whose trajectories and fixed points encode properties of both the model and the data.
- We demonstrate that most neural mappings are contractive, leading naturally to the emergence of fixed points and attractors in the latent space.
- We empirically connect attractors to the network's memorization and generalization regimes, showing how they evolve during training.
- We show experimentally that vision foundation models can be probed in a data-free manner: initializing with noise, we recover attractors that reveal semantic information embedded in the model's weights.
- Finally, we show that trajectories in the latent vector field can capture the learned data distribution and serve as a signal for detecting distribution shifts.

## 2 METHOD

**Notation and background** We consider neural models $F$ as compositions of encoder-decoder modules, defined as $F_\Theta = D_{\theta_2} \circ E_{\theta_1}$ parametrized by $\Theta = [\theta_1, \theta_2]$. The encoder $E_{\theta_1}$ maps inputs $\mathbf{x} \sim p(\mathbf{x})$ supported on $\mathcal{X} \subset \mathbb{R}^m$ to a typically lower-dimensional space $\mathcal{Z} \subset \mathbb{R}^k$, and the decoder $D_{\theta_2}$ reconstructs the input. The model is trained to minimize the mean squared error $\mathcal{L}_{MSE}$:

$$\mathcal{L}_{MSE}(\mathbf{x}) = \sum_{\mathbf{x} \in X} \|\mathbf{x} - F_\Theta(\mathbf{x})\|_2^2 + \lambda \mathcal{R}(\Theta), \tag{1}$$

where $\mathcal{R}$ is either an explicit or implicit regularization term encouraging contractive behavior in $F$.

We posit that minimizing this objective leads to a reduction in the spectral norm of the Jacobian, $\|J_F(\mathbf{x})\|_\sigma$ for $\mathbf{x} \sim p(\mathbf{x})$. Examples of regularization include weight decay, where $\mathcal{R} = \|\Theta\|_2^F$, or data augmentation by sampling transformations $T \sim p(T)$ and minimizing:

$$\mathcal{L}_{MSE}(\mathbf{x}) = \sum_{\mathbf{x} \in \mathcal{X}} \|\mathbf{x} - F(\mathbf{x})\|_2^2 + \sum_{T \in p(T)} \|\mathbf{x} - F(T\mathbf{x})\|_2^2. \tag{2}$$

Such augmentations include additive Gaussian noise in denoising AEs (Vincent et al., 2008) and input masking in masked AEs (He et al., 2022). Another form of contractive pressure is the bottleneck dimension $k = \dim(\mathcal{Z})$, which places a hard upper bound on the rank of the encoder Jacobian, $\text{rank}(J_E) <= k$ and, by the chain rule, constrains the full model Jacobian $J_F(\cdot) = J_E(\cdot) \odot J_D(\cdot)$. Table 3 (Appendix) lists many AE variants, including denoising AEs (DAEs) (Vincent et al., 2008), sparse AEs (SAEs) (Ng et al., 2011), variational AEs (VAEs) (Kingma et al., 2013), and others (Rifai et al., 2011; Alain and Bengio, 2014; Gao et al., 2024), highlighting how their objectives promote local contractive behavior around training data.

Given a possibly pretrained AE model, we define the map $f(\mathbf{z}) = E \circ D(\mathbf{z})$ and study its repeated application $f(\ldots f(f(\mathbf{z})))$, which can be modeled as a differential equation.

### 2.1 THE LATENT DYNAMICS OF NEURAL MODELS

Given a sample $\mathbf{z} \in \mathcal{Z} \subseteq \mathbb{R}^k$, we study the effect of repeatedly applying the map $f$, i.e., $f \circ f \circ \ldots f(\mathbf{z})$. By introducing a discrete time parameter $t$, this iterative process defines the discrete ODE:

$$\begin{cases} \mathbf{z}_{t+1} = f(\mathbf{z}_t) \\ \mathbf{z}_0 = \mathbf{z} \end{cases} \tag{3}$$

which discretizes the following continuous differential equation:

$$\frac{\partial \mathbf{z}}{\partial t} = f(\mathbf{z}) - \mathbf{z} \tag{4}$$

In Eq. 3, the map $f$ defines the pushforward of a *latent vector field* $V : \mathbb{R}^k \mapsto \mathbb{R}^k$, tracing nonlinear trajectories in latent space (Fig. 1). A natural question is whether the ODE has well-defined and unique solutions. By the Banach fixed-point theorem, this holds if and only if $f$ is Lipschitz-continuous with Lipschitz constant $C$<1.

**Definition 1.** *A function $f : \mathcal{Z} \mapsto \mathcal{Z}$ is Lipschitz-continuous if there exists a constant $C$ s.t. for every pair of points $\mathbf{z_1}, \mathbf{z_2}$:*

$$d(f(\mathbf{z_1}), f(\mathbf{z_2}))_{\mathcal{Z}} \leq C \, d(\mathbf{z_1}, \mathbf{z_2})_{\mathcal{Z}} . \tag{5}$$

*When $C < 1$, $f$ is called* contractive, *for a given metric $d$ on $\mathcal{Z}$.*

For any contractive map, Eq. 3 admits fixed-point solutions, i.e., repeatedly applying the map $f$ will converge to a unique solution $\mathbf{z}^*$ satisfying $\mathbf{z}^* = f(\mathbf{z}^*)$. The fixed points $\mathbf{z}^*$ can act as attractors, capturing and summarizing the system's long-term dynamics.

**Definition 2.** *A fixed point $\mathbf{z} = f(\mathbf{z})$ is an attractor of a differentiable map $f$ if all eigenvalues of the Jacobian $J$ of $f$ at $\mathbf{z}$ are strictly less than one in absolute value.*

When $f$ is nonlinear, the ODE in Eq. 3 can have multiple solutions depending on the initial conditions $\mathbf{z_0}$. The previous definition allows for a definition of Lipschitz continuity, which is inherently local:

**Definition 3.** *Let $f : \mathcal{Z} \mapsto \mathcal{Z}$ be differentiable and $C$-Lipschitz continuous. Then the Lipschitz constant $C$ is given by: $C = sup_{\mathbf{z} \in \mathcal{Z}} \|J_f(\mathbf{z})\|_\sigma$ , where $J_f(\mathbf{z})$ is the Jacobian of $f$ evaluated at $\mathbf{z}$ and $\| \cdot \|_\sigma$ is the spectral norm.*

The set of initial conditions $\mathbf{z}_0$ leading to the same attractor $\mathbf{z}^*$, is denoted as *basin of attraction*.

**Why are neural mappings contractive in practice?** We argue that mappings learned by neural AEs and their variants tend to be *locally contractive*, i.e.,their Jacobians have small eigenvalues near training examples. This behavior emerges naturally from several explicit and implicit inductive biases present in modern training pipelines. Below, we outline the main factors promoting contractive behavior:

- **Initialization bias.** Standard initialization schemes (LeCun et al., 2002; Glorot and Bengio, 2010; He et al., 2015) are designed to preserve activation variance at the start of training to avoid vanishing or exploding gradients. Theoretical arguments in support assume i.i.d. inputs and weights, and are often tied to specific activation functions (e.g., ReLU in (He et al., 2015)). However, real-world training data is typically correlated, and architectural features like residual connections (He et al., 2016) break weight independence. As a result, networks often exhibit a bias toward mappings that are globally expansive or contractive, empirically skewed toward the latter (Poole et al., 2016).

  We illustrate this empirically in Figure 6 (Appendix), showing that various vision backbones exhibit contractive behavior at initialization. Figure 3a (left) shows a 2D latent vector field which is globally contractive at initialization towards a single attractor at the origin. This contractive behavior also holds in higher dimensions (see the number of attractors at epoch 0 in Figure 3c).

- **Explicit regularization.** Common regularization methods like weight decay (D'Angelo et al., 2024) encourage contraction by penalizing the norm of model parameters. In the linear case, minimizing parameter norms directly reduces the Jacobian's spectral norm, making the map contractive. While this link is less direct for nonlinear models, the effect due to weight decay persists in practice. Regularization is often integrated into optimizers used in large-scale models (Loshchilov, 2017). Additional architectural constraints, such as small bottleneck dimensions, also limit the rank of $J_F(\mathbf{z})$. Soft constraints include KL divergence in VAEs or sparsity penalties in SAEs (see Table 3 in the Appendix for a complete overview).

- **Implicit regularization.** Data augmentations introduce local perturbations around training examples, effectively defining a neighborhood structure in input space. For instance, Gaussian noise in DAEs or masking in MAEs perturbs inputs along specific directions. These augmentations implicitly regularize the Jacobian $J_f(\mathbf{x})$ by penalizing sensitivity to those perturbations. Unlike parametric regularization, this effect is inherently local and nonparametric.

## 3 THEORETICAL REMARKS

In this section, we study the behavior of the latent vector field, focusing on its trajectories (Section 3.1) as well as its fixed points and attractors (Section 3.2), when they exist. Formal statements and proofs of all theorems and propositions are provided in Appendix A.

### 3.1 TRAJECTORIES OF THE LATENT VECTOR FIELD

Prior work (Miyasawa et al., 1961; Robbins, 1992; Alain and Bengio, 2014) has shown that, under ideal conditions, the residual of a denoising AE trained with noise variance $\sigma^2$ approximates the score function, i.e., the gradient of the log-density of the data, as $\sigma \to 0$. Interestingly, Alain and Bengio (2014) shows a connection between denoising and contractive AEs, where the Jacobian norm of the input-output mapping is explicitly penalized.

We build on this by observing a more general phenomenon: when an autoencoding map is locally contractive relative to a chosen neighborhood structure (e.g., Gaussian noise, masking) and sufficiently approximates the input distribution $p(\mathbf{x})$, the induced latent vector field pushes points in the direction of the score of the corresponding prior in latent space.

Informally, this implies that the vector field acts to nonlinearly project samples toward regions of high probability on the data manifold.

**Theorem 1** (informal, proof in Appendix A.1). *Let $F$ be a trained autoencoder and let $q(\mathbf{z}) = \int p(\mathbf{x}) q(\mathbf{z}|\mathbf{x}) d\mathbf{x}$ be the marginal distribution induced in latent space. Assume $q(\mathbf{z})$ is smooth and that there exists an open neighborhood $\Omega \supseteq \operatorname{supp} q$ and a constant $L < 1$ such that $\sup_{\mathbf{z} \in \Omega} \left\| J_f(z) \right\|_{\sigma} \leq L$. Then, latent dynamics $f(\mathbf{z}) - \mathbf{z}$ in $\Omega$ is locally proportional to the score function $\nabla_{\mathbf{z}} \log q(\mathbf{z})$.*

This result establishes a general link between the vector field's trajectories and the score function, under the assumption of local contractivity. In practice, neural networks often strike a balance between the reconstruction loss (Eq. 1) and regularization on the Jacobian, which contributes to the emergence of attractors in the latent space.

An important implication of Theorem 1 is that integrating the vector field $f(\mathbf{z}) - \mathbf{z}$ effectively estimates the log-density, i.e., $\int_{\mathbf{z}} \nabla \log q(\mathbf{z}) dz = \log q(\mathbf{z}) + C$ in unbounded domains. If the Jacobian $J_f$ is symmetric (e.g., in AEs with tied weights (Alain and Olivier, 2013)), then the latent vector field is *conservative* and corresponds to the gradient of a potential $V_f = \nabla E$. In this case, the AE defines an energy-based model with $q(\mathbf{z}) = e^{-E(\mathbf{z})}$ (Zhai et al., 2016; Song and Kingma, 2021). In the general (non-conservative) case, we show empirically in Section 4.2 that the trajectories still reflect the learned prior distribution.

However, attractors are generally not accessible solving the fixed-point equation with first order methods, unless initialized very close to them. The following result formalizes this:

**Proposition 3.1** (informal, proof in Appendix A.2). *Iterations of the map $f$ in Eq. 3 correspond to gradient descent on a potential $L(\mathbf{z})$ (i.e., $f \approx \mathbf{z} - \alpha L(\mathbf{z})$) if and only if $f$ is an isometry (its eigenvalues are near 1), or the dynamics occur near attractors, where $J_f$ vanishes.*

This underscores the role of higher-order dynamics in the vector field: repeated applications of $f$ trace complex, nonlinear trajectories that can escape spurious or unstable fixed points.

### 3.2 CHARACTERIZING THE ATTRACTORS OF THE LATENT VECTOR FIELD

In this section, we aim to characterize what the fixed point solutions of Eq. 3 represent. In Appendix A.4 we consider the two cases of linear and homogeneous networks as examples, to build intuition on the characterization of attractors and latent vector field.

**Between memorization and generalization.** Our goal is to characterize the properties of the latent vector field and the learned attractors in the general setting. We argue that different models can fit training points reaching similar low loss, but interpolate *differently* outside the training support, depending on the regularization strength and the amount of overparametrization.

In this context, a key question arises: what information do the attractors $\mathbf{z}^*$ encode? We argue that neural models lie in a *spectrum* between memorization and generalization, depending on the strength

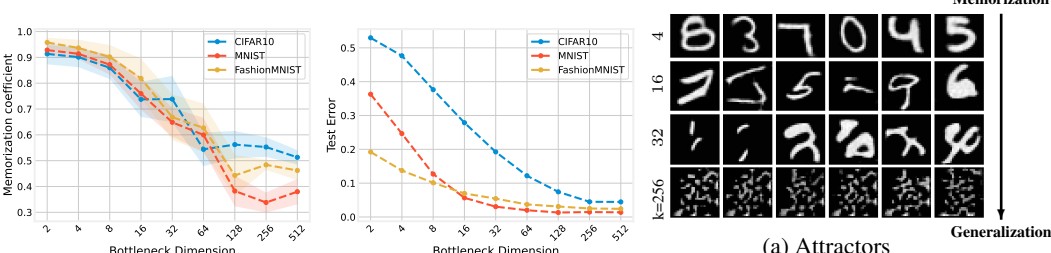

Figure 2: *Memorization vs Generalization*. Attractors memorize the training data as a function of the rank of $J_f(\mathbf{z})$ by adjusting the bottleneck dimension $k$ *(left)* which is inversely proportional to the amount of generalization attained by the model *(center)*; On the *right* we show example of decoded attractors transitioning from a strong memorization model *(first row)* to good generalization *(last row)*.

of the regularization term in Eq. 1. We hypothesize that attractors fully characterize where a model falls on this spectrum. To do so, we first connect attractors to the notion of generalization via the following proposition:

**Proposition 3.2** (Informal, proof in Appendix A.3). *Let $\mathbf{Z}^*$ be a dictionary of attractors of $f = E \circ D$ in a neighborood $\Omega$ of the latent space, and let $\Pi(\mathbf{z})$ denote the projection onto the nearest attractor to a latent code $\mathbf{z} = E(\mathbf{x})$. If $D$ is $L_D$-Lipschitz on $\Omega$, then for any test point $\mathbf{x}$:*

$$\|\mathbf{x} - F(\mathbf{x})\|_2^2 \;\leq\; \underbrace{\|\mathbf{x} - D(\Pi(E(\mathbf{x})))\|_2^2}_{\text{error to prototype}} + \underbrace{L_D^2 \, \|E(\mathbf{x}) - \Pi(E(\mathbf{x}))\|_2^2}_{\text{coverage error}}.$$

In short, attractors define a dictionary for the data: when they coincide with training points, prototype error on these points vanishes but coverage is narrow (memorization regime), whereas generalization requires attractors that both cover the latent space and serve as good prototypes for unseen data.

We consider attractors as *representations summarizing the information stored in the weights of the network*. This interpretation is in line with the convergence of paths in the vector field to modes of the learned distribution of Theorem 1, and can be easily seen in the case of memorization.

**Extreme overparametrization case.** When the capacity of a network exceeds the number of training examples by far, AEs enter an overfitting regime leading to data memorization (Zhang et al., 2019; Radhakrishnan et al., 2020; Jiang and Pehlevan, 2020; Kadkhodaie et al., 2023). The network in this case learns a constant function or an approximation thereof, which can be retrieved through the iterations of Eq. 3.

### 3.2.1 THE ROLE OF REGULARIZATION

We empirically demonstrate that the balance between memorization and generalization is governed by the strength of regularization. In Figure 2, we show how increasing regularization on the Jacobian $J_f(\mathbf{z}^*)$ drives the model from a memorization regime, where many training examples are stored as attractors, toward generalization. We remark that this memorization arises in an *over-regularized* regime, as opposed to the *overfitting* regime of extremely overparametrized networks.

**Setting.** We trained 30 convolutional AEs on the `CIFAR`, `MNIST`, and `FashionMNIST` datasets, varying the bottleneck dimension $k$ from 2 to 512. This acts as a hard regularizer on $J_f(\mathbf{z}^*)$, by constraining its rank to be $\leq k$. We compute attractors $\mathbf{Z}^*$ from elements of the training set $\mathcal{X}_{train}$ by iterating $f$ till convergence. We measure the degree of memorization by defining a *memorization coefficient*, which is given by the cosine similarity of each decoded attractor $\mathbf{x}^* = D(\mathbf{z}^*)$ to its closest point in the training set, $\mathrm{mem}(\mathbf{z}^*) = \max_{x \in \mathcal{X}_{train}} \cos(D(\mathbf{z}^*), \mathbf{x})$ and we report the mean and standard deviation over attractors. We measure generalization by simply reporting the error on the test set. In Figure 13 in the Appendix, we also report the rank explaining 90% of the variance of the matrix of decoded attractors $\mathbf{X}^*$, showing that attractors corresponding to good generalization models span more directions in the input space. For additional information on the model, hyperparameters, and settings, we refer to Appendix D.

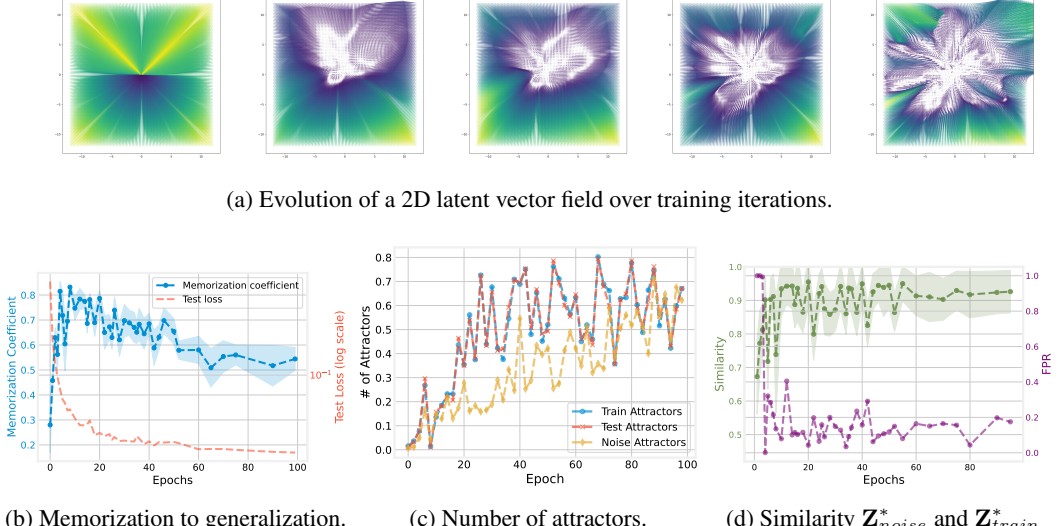

(a) Evolution of a 2D latent vector field over training iterations.

(b) Memorization to generalization.  (c) Number of attractors.  (d) Similarity $\mathbf{Z}^*_{noise}$ and $\mathbf{Z}^*_{train}$

Figure 3: *Latent vector field dynamics. (a)* The 2D vector field ($k = 2$) expands from a single attractor, eventually stabilizing and over-fitting because of capacity limits. *Bottom:* Evolution of larger capacity AEs ($k = 128$) across training. *(b)* Throughout training, the network first memorizes the data with a high memorization coefficient (in blue) and then generalizes, achieving a low test error (red). *(c):* Evolution of attractor count for training (blue), test (red), and noise (yellow) samples; *(d)* Attractors computed from training and from gaussian noise converge during training (*green*), while the separability of the trajectories (measured as FPR95, the lower the better) increase (*purple*).

**Analysis of results.** In Figure 2 on the left, we observe that the more regularized networks ($k$ from 2 to 16) tend to memorize data, trading off generalization performance (middle plot). We remark that this kind of memorization is not due to overfitting, as was shown in (Kadkhodaie et al., 2023) for diffusion models, but it happens in the underfit regime, due to the strong regularization constraint on $J_f(\mathbf{x})$. We show in Figure 14 in the Appendix results in the overfitting regime, by training the models with different sample sizes and observing a similar pattern.

> **Takeaway.** Attractors capture the interplay between generalization and memorization of neural models, which corresponds to the trade-off between the reconstruction performance and regularization term of the AE model.

### 3.2.2 MEMORIZATION AND GENERALIZATION ACROSS TRAINING

In the experiment in Figure 3 we show that a similar transition from a memorization regime to generalization occurs across training.

**Setting.** We monitor the latent vector field and attractors statistics across the training dynamics of a convolutional AE trained on MNIST with bottleneck dimension $k = 128$ (bottom row plots in the Figure). To show qualitatively the evolution of a latent vector field, we also plot it across training epochs for AEs with bottleneck dimension $k = 2$ in Figure 3a.

**Analysis of results.** In Figure 3b we show the transition from memorization, occurring at the first epochs of training, to generalization, by plotting the memorization coefficient and the test error across training, observing a trade-off between the two. In the center plot, we plot the fraction of distinct attractors, computed from 5000 random elements respectively from the $P_{train}, P_{test}$ and $\mathcal{N}(0, I)$, where we consider two attractors equal if $\cos(\mathbf{z}^*_1, \mathbf{z}^*_2) > 0.99$. Initially, the model converges to a single attractor (as also seen in the 2D example in Figure 3a). Over time, the number of attractors increases, stabilizing for training and test data in tandem with the test loss, while attractors from noise inputs converge more slowly. The right plot shows the Chamfer symmetric similarity between attractors from the training and noise distributions, which increases over training as the two distributions of attractors match.

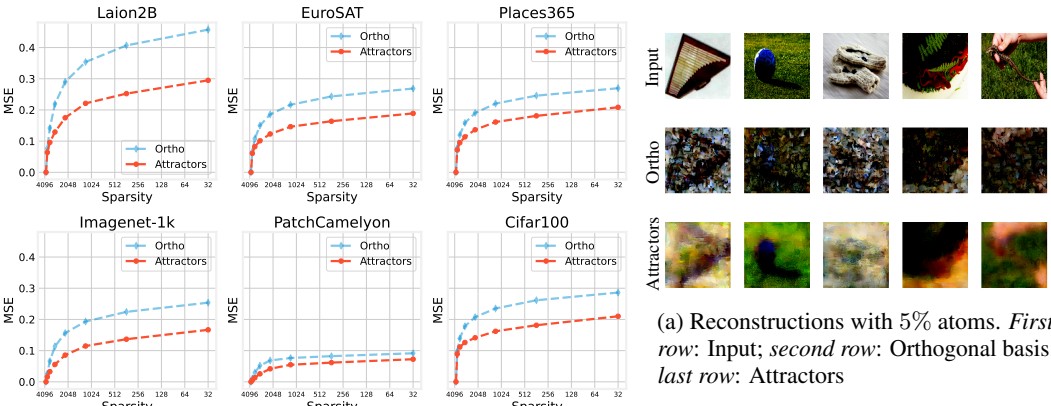

(a) Reconstructions with $5\%$ atoms. *First row*: Input; *second row*: Orthogonal basis; *last row*: Attractors

Figure 4: *Data-free weight information probing of Stable Diffusion model.* We plot the error (MSE) vs sparsity (number of atoms) used to reconstruct samples from diverse dataset respectively from (i) an orthonormal random basis of the latent space (*blue*); (ii) attractors computed from gaussian noise (*red*), showing that attractors consistently reconstructs samples better on all datasets. (*Right*) Reconstructions using $5\%$ of the atoms on `ImageNet`

Importantly, while attractors derived from noise and training data become increasingly similar over the course of training, the *trajectories* leading to them differ significantly. To quantify this, we compute the False Positive Rate at $95\%$ True Positive Rate (FPR95) scores for distinguishing trajectories originating from noise versus those from training data: FPR95 measures what percentage of noise latent trajectories are falsely classified as training trajectories, setting a threshold such that $95\%$ of training trajectories are classified correctly. We define the trajectory score as $\text{score}(\mathbf{z}) = \frac{1}{N}\sum_{\mathbf{z}_i \in \pi(\mathbf{z})} d(\mathbf{z}_i, \mathbf{Z}^*_{train})$ where $\pi(\mathbf{z}) = [\mathbf{z}_0, ..., \mathbf{z}_N]$ is the trajectory. The FPR95 decreases sharply during training, showing that the network learns to separate the two types of trajectories.

> **Takeaway.** As the latent vector field evolves during training, the model transitions from memorization to generalization, forming similar attractors from different input distributions, while retaining information of the source distribution in the latent trajectories.

## 4    EXPERIMENTS ON VISION FOUNDATION MODELS

In this section we demonstrate the existence and applicability of the latent dynamics on vision foundation models, including the AE backbones of the Stable Diffusion model (Rombach et al., 2022) and vision transformer masked AEs (He et al., 2022), showing; *(i)* how information stored in the weights of a pretrained model can be recovered by computing the attractors from noise *(ii)* how trajectories in the latent dynamics are informative to characterize the learned distribution and detect distribution shifts.

### 4.1    DATA FREE WEIGHT PROBING

**Setting.** In this experiment, we investigate how much information stored in a neural network's weights can be recovered purely from attractors computed on Gaussian noise, *without* access to any input data. We focus on AE component of Stable Diffusion (Rombach et al., 2022) pretrained on the large scale `Laion2B` dataset (Schuhmann et al., 2022).

We sample $\mathbf{Z}_n \sim \mathcal{N}(\mathbf{0}, I)$ and compute the corresponding attractors as solutions to $f(\mathbf{Z}^*_n) = \mathbf{Z}^*_n$. We generate $N = 4096$ such points, matching the latent dimensionality $k$ of the model. We evaluate the reconstruction performance on 6 diverse datasets `Laion2B` (Schuhmann et al., 2022), `Imagenet` (Deng et al., 2009), `EuroSAT` (Helber et al., 2019), `CIFAR100` (Krizhevsky et al., 2009), `PatchCamelyon` (Litjens et al., 2018), and `Places365` (Zhou and Paffenroth, 2017). These datasets span general, medical, and satellite image domains.

For each dataset, we randomly sample 500 test examples and reconstruct them in latent space using Orthogonal Matching Pursuit (OMP) (Mallat and Zhang, 1993), with varying sparsity levels (i.e.,

| Method | SUN397 | | Places365 | | Texture | | iNaturalist | |
|---|---|---|---|---|---|---|---|---|
| | FPR ↓ | AUROC ↑ | FPR ↓ | AUROC ↑ | FPR ↓ | AUROC ↑ | FPR ↓ | AUROC ↑ |
| d(Attractors) | **29.60** | **91.20** | **29.95** | **90.99** | **25.85** | **92.63** | **29.85** | **91.29** |
| KNN | 100.00 | 42.59 | 100.00 | 32.36 | 34.50 | 89.41 | 86.35 | 68.60 |

(a) *FPR and AUROC scores*

(b) *KNN baseline*  (c) *Latent trajectories*

Figure 5: *Trajectories in the latent vector field characterize distribution shifts* We measure out-of-distribution detection performance on `ViTMAE`: On the *left* we report scores for 4 different datasets, highly outperforming the KNN baseline. On the *right*, histograms of scores on the INaturalist dataset, demonstrating much better separability between in-distribution and out-of-distribution when probing latent trajectories distances to in-distribution attractors (c), as opposed to measure distances to in-distribution features (b).

number of atoms used). Reconstructions are decoded via $D$, and performance is compared against a baseline reconstruction using the initial orthogonal samples $\mathbf{Z}_0^*$, which, by design, fully span the latent space $\mathcal{Z}$.

**Analysis of results.** In Figure 4, we plot the error as a function of the number of elements (atoms) chosen to reconstruct the test samples using OMP. We compare by building a random orthogonal basis sampled in the latent space. On all considered datasets, noise attractors recover test samples with lower reconstruction error, representing a better dictionary of signals. In Figure 4a, we show qualitative reconstruction using only 5 % of the atoms. In Figures 7, 8, 9 in the Appendix, we include additional qualitative evidence of this phenomenon, visualizing reconstructions of random samples of the datasets as a function of the number of atoms used. In Appendix B, we report additional results on variants of the AE of different sizes.

> **Takeaway.** Attractors of foundation models computed from noise can serve as a dictionary of signals to represent diverse datasets, demonstrating that it is possible to probe the information stored in the weights of foundation models in a black box way, without requiring any input data.

## 4.2 LATENT TRAJECTORIES CHARACTERIZE THE LEARNED DISTRIBUTION

In the following experiment, we test the hypothesis that trajectories are informative on the distribution learned by the model, testing how well Theorem 1 holds in practice. Our goal is to evaluate whether these trajectories can be used to detect distribution shifts in the input data. To classify a sample as out-of-distribution (OOD), we focus on two key questions: (i) does the sample trajectory converge to one of the attractors of the training data, i.e., does it *share the same basin* of attraction? (ii) If so, *how fast* does it converge?

Notably, an OOD sample may still lie within a shared attractor basin. To capture both scenarios, we track the distance from each point along a test sample's trajectory to the nearest attractor from the training set. In the former case, we expect that an OOD sample converges faster, while in the latter case, the distance term to the attractors will dominate the score.

**Setting.** We test this hypothesis using the `ViT-MAE` (He et al., 2022) architecture pretrained on `ImageNet`. We sample 2000 training images and compute their attractors, stopping when convergence reaches a tolerance of $10^{-5}$ or a maximum number of iterations. We test on samples from `SUN397` (Xiao et al., 2016), `Places365` (Zhou and Paffenroth, 2017), `Texture` (Cimpoi et al., 2014), and `iNaturalist` (Van Horn et al., 2018), standard benchmarks for OOD detection (Yang et al., 2024). We report two metrics: FPR95, and Area Under the Receiver Operating Characteristic Curve (AUROC). FPR95 here measures what percentage of OOD data we falsely classify as ID, setting a threshold s.t. 95% of ID data is correctly classified.

**Analysis of results.** In Figure 5 we report OOD detection performance by using the following score function: for a test sample $\mathbf{z}_{test}$ we compute its trajectory $\pi_{\mathbf{z}_{test}} = [\mathbf{z}_0, ..., \mathbf{z}_N^*]$ towards attractors and compute the distance of $\pi_{\mathbf{z}_{test}}$ to the set of training attractors $\mathbf{Z}_{train}^*$, i.e. $\text{score}(\mathbf{z}) = d(\pi_{\mathbf{z}_{test}}, \mathbf{Z}_{train}^*)$. As a distance function, we employ Euclidean distance, and we aggre-

gate the score by computing the mean distance over the trajectory $\text{score}(\mathbf{z}) = \frac{1}{N} \sum_i d(\mathbf{z}_i, \mathbf{Z}^*_{train})$. We compare with a $K$-Nearest neighbor baseline (adapted from Sun et al. (2022)), where the score for a test sample is obtained by taking the mean distance over the $K$-NN on the training dataset, where $K = 2000$ in the experiments. The score proposed demonstrates how informative the latent vector field is on the training distribution. In Figure 5 on the right, we show histograms of scores for the distance to attractors and the KNN, showing again that the former method is able to tell apart in-distribution and out-of-distribution data correctly.

> **Takeaway.** Trajectories in the latent vector field characterize the source distribution and are informative to detect distribution shifts.

## 5   RELATED WORK

**Memorization and generalization in neural networks (NNs).** NNs exhibit a rich spectrum of behaviors between memorization and generalization, depending on model capacity, regularization, and data availability (Arpit et al., 2017; Zhang et al., 2021; Power et al., 2022). In the case of extreme overparametrization, namely networks trained on few data points, it has been shown experimentally in (Radhakrishnan et al., 2020; Zhang et al., 2019) and theoretically for sigmoid shallow AEs in (Jiang and Pehlevan, 2020) that AEs can memorize examples and implement associative memory mechanisms. (Alain and Bengio, 2014; Vincent, 2011), A similar phenomenon has been observed in diffusion models in (Somepalli et al., 2023; Dar et al., 2023) and analyzed in (Kadkhodaie et al., 2023) Similarly, non gradient-based approaches such as Hopfield networks and their modern variants (Hopfield, 1982; Ramsauer et al., 2020) extend classical attractor dynamics to neural systems that interpolate between memory-based and generalizing regimes. In our work, we show that AEs fall in general in the spectrum between memorization and generalization, depending on inductive biases that enforce contraction.

**Contractive neural models.** Different approaches have been proposed to regularize NNs in order to make them smoother and more robust to input perturbations and less prone to overfitting. Many of these regularization techniques either implicitly or explicitly promote contractive mappings in AEs: for example, sparse AEs (Ng et al., 2011; Gao et al., 2024), their denoising variant (Vincent et al., 2008), and contractive AEs (Rifai et al., 2011; Alain and Bengio, 2014) losses enforce learned maps to be contractive through the loss. Regularization strategies such as weight decay (Krogh and Hertz, 1991) favor as well contractive solutions for neural models, and they are incorporated in standardized optimizers such as AdamW (Loshchilov, 2017). All these approaches enforce either directly or indirectly the existence of fixed points and attractors in the proposed latent vector field representation.

**Neural networks as dynamical systems.** Distinct lines of work have interpreted neural networks as dynamical systems. Neural ODEs (Chen et al., 2018) view depth as continuous time and model hidden states via differential equations, while deep equilibrium models (Bai et al., 2019) characterize predictions as fixed points of implicit dynamics. Closer to our work, Radhakrishnan et al. (2020) interpret overparameterized autoencoders as dynamical systems acting in the input space, implementing associative memory. In contrast, we show that *any* autoencoder induces a *latent* vector field, and we link its properties to generalization, memorization, and the characterization of the learned distribution.

**Nonlinear operators spectral analysis.** In the context of image processing and 3D graphics, previous work has inspected generalization of spectral decompositions to nonlinear operators (Bungert et al., 2021; Gilboa et al., 2016; Fumero et al., 2020), focusing on one homogeneous operator. In our work, fixed points of Eq. 3 can be interpreted as the decomposition of the NN into a dictionary of signals.

## 6   CONCLUSIONS AND DISCUSSION

In this work we proposed to represent neural AEs as vectors fields, implicitly defined by iterating the autoencoding map in the latent space. We showed that *(i)* attractors in the latent vectors field exists in practice due to inductive biases in the training regime which enforce local contractions; *(ii)* they retain key properties of the model and the data, linking to memorization and generalization regimes of the model; *(iii)* knowledge stored in the weights can be retrieved without access to input data in vision foundation models; *(iv)* paths in the vector field inform on the learned distribution and its shifts.

### 6.1 LIMITATIONS AND FUTURE WORKS

**Generalizing to arbitrary models.** Eq. 3 cannot be directly generalized to model trained with discriminative objectives such as a deep classifiers, or self supervised models, as the network is not invertible. However we note that (i) our theory holds in general for any self map which is locally contractive (ii) the vector field is still defined in the output space, and can be simply obtained by computing the residual $F(\mathbf{x}) - y$ and in the neighborhood of an attractor, the relation in proposition 3.1 can still hold for different objectives. One idea is to train a decoder on top of the frozen encoder-only model. We give preliminary evidence of the existence of latent vector fields in self-supervised models and in next token predictors (e.g. LLMs) in Appendix B.8, showing that extending our framework to arbitrary models holds promise. An alternative intriguing idea is the one to train a *surrogate AE* model in the latent space of the model of interest, which would be agnostic from the pretraining objective. Sparse AEs for mechanistic interpretability of large language models (LLMs) (Gao et al., 2024) fit in this category. Analyzing the associated latent vector field can shed light on features learned by SAEs and biases stored in their weights.

**Learning dynamics.** Characterizing how attractors forms during training, under which conditions noise attractors converge to the training attractors, holds promise to use the proposed representation to study the learning dynamics of neural models to inspect finetuning of AE modules, such as low-rank adapters (Hu et al., 2022) and double descent (Nakkiran et al., 2021).

**Alignment of latent vector fields.** Finally, following recent findings in representation alignment (Moschella et al., 2022; Fumero et al., 2024; Huh et al., 2024) inspecting how latent vector fields of networks trained are related is an open question for future works, to possibly use this representations to compare different neural models.

## ACKNOWLEDGMENTS

This work is supported by the MUR FIS2 grant n. FIS-2023-00942 "NEXUS" (cup B53C25001030001), and partly by Sapienza University of Rome via the Seed of ERC grant "MINT.AI" (cup B83C25001040001).This research was also funded in whole or in part by the Austrian Science Fund (FWF) 10.55776/COE12. For open access purposes, the author has applied a CC BY public copyright license to any author accepted manuscript version arising from this submission. MF is supported as well by the MSCA IST-Bridge fellowship which has received funding from the European Union's Horizon 2020 research and innovation program under the Marie Skłodowska-Curie grant agreement No 101034413. M.F. thanks C.Domine, V.Maiorca, I.Cannistraci, R.Cadei, B.Demirel, S.Vedula for insightful discussions.

## ETHICS STATEMENT

This work is primarily methodological and theoretical, focusing on the dynamics of autoencoder networks. We do not foresee any direct negative societal impacts. All datasets used are publicly available benchmark datasets (e.g., MNIST, CIFAR, Imagenet, Laion2B) or openly released via HuggingFace (Lhoest et al., 2021). No sensitive, private, or personally identifiable information is involved. Our experiments are designed for scientific analysis only, and no human subjects or protected groups are included. While the methodology in Section 4.1 provides tools to probe information about trained data in autoencoders foundation models, it does not directly improve fairness, robustness, or safety. In principle, insights into memorization could be misused to develop algorithms to recover information from pretrained models in unintended ways. However, our study is limited to publicly available models and datasets, and we emphasize that the proposed techniques should be applied responsibly in research contexts. We hope our findings contribute to a better understanding of neural representations and their generalization/memorization properties, which may inform the design of more transparent and robust models. We provide details and hyperparameters for each experiment for reproducibility and formal statements and proof of our theoretical results in the Appendix.

## REPRODUCIBILITY STATEMENT

All formal statements and proofs of theorems are reported in Appendix A. We implement all our experiments using PyTorch (Paszke et al., 2019) and HuggingFace libraries Lhoest et al. (2021); Wolf et al. (2020). We experiment solely with openly available models and datasets available on HuggingFace Hub. We will open-source our codebase for all the experiments upon acceptance. Unless otherwise noted in the experiment-specific sections, all hyperparameters are listed in Section D.

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

# A  PROOFS AND DERIVATION

## A.1  THEOREM 1

We report here a detailed formulation of Theorem 1 alongside its proof.

**Assumption A.1** (Latent marginal). Let $p_{\text{data}}(\mathbf{x})$ be the data distribution and $q_\phi(\mathbf{z} \mid \mathbf{x})$ an encoder mapping inputs $\mathbf{x} \in \mathbb{R}^n$ to latent codes $\mathbf{z} \in \mathbb{R}^d$. The latent marginal density is defined as:

$$q(\mathbf{z}) = \int p_{\text{data}}(\mathbf{x}) \, q_\phi(\mathbf{z} \mid \mathbf{x}) \, d\mathbf{x}.$$

We assume that $q(\mathbf{z})$ is continuously differentiable, and that its gradient $\nabla \log q(\mathbf{z})$ and Hessian $\nabla^2 \log q(\mathbf{z})$ are well-defined and continuous on an open domain containing $\Omega \subseteq \mathbb{R}^d$.

**Assumption A.2** (Fixed-point manifold). Let $E : \mathbb{R}^n \to \mathbb{R}^d$ be an encoder and $D : \mathbb{R}^d \to \mathbb{R}^n$ a decoder, both continuously differentiable. Define the composite map

$$f(\mathbf{z}) = E(D(\mathbf{z})) \in \mathbb{R}^d.$$

We define the fixed-point manifold of $f$ as

$$\mathcal{M} = \left\{ \mathbf{z} \in \mathbb{R}^d \ : \ f(\mathbf{z}) = \mathbf{z} \right\}.$$

**Assumption A.3** (Local contraction). There exists an open, convex set $\Omega \subseteq \mathbb{R}^d$, with $\mathcal{M} \subseteq \Omega$, and a constant $0 < L < 1$, such that:

$$\sup_{\mathbf{z} \in \Omega} \|J_f(\mathbf{z})\|_\sigma \le L,$$

where $J_f(\mathbf{z}) \in \mathbb{R}^{d \times d}$ denotes the Jacobian of $f$ at $\mathbf{z}$, and $\| \cdot \|_\sigma$ denotes the spectral norm (i.e., largest singular value). Therefore, $f$ is a contraction mapping on $\Omega$.

**Assumption A.4** (Training optimality with Jacobian regularization). Let $f(\mathbf{z}) = E(D(\mathbf{z}))$ be as above, with $E, D$ trained to minimize the objective:

$$\mathbb{E}_{\mathbf{x} \sim p_{\text{data}}} \left[ \|D(E(\mathbf{x})) - \mathbf{x}\|^2 + \lambda \|J_f(E(\mathbf{x}))\|_F^2 \right] \quad \text{for } \lambda > 0.$$

For example, assume the encoder is deterministic, i.e., $q_\phi(\mathbf{z}|\mathbf{x}) = \delta(\mathbf{z} - E(\mathbf{x}))$, so the latent marginal satisfies:

$$q(\mathbf{z}) = \int p_{\text{data}}(\mathbf{x}) \, \delta(\mathbf{z} - E(\mathbf{x})) \, d\mathbf{x}.$$

Then $q(\mathbf{z})$ is supported and concentrated on the fixed-point manifold $\mathcal{M}$, and $f$ is locally contractive around $\mathcal{M}$.

**Lemma A.5** (Directional ascent of the residual field). *Under the above assumptions, for every* $\mathbf{z} \in \Omega \setminus \mathcal{M}$, *define the residual field:*

$$\mathbf{v}(\mathbf{z}) := f(\mathbf{z}) - \mathbf{z}.$$

*Then the directional derivative of* $\log q$ *in direction* $\mathbf{v}(\mathbf{z})$ *is strictly positive:*

$$\langle \nabla \log q(\mathbf{z}), \mathbf{v}(\mathbf{z}) \rangle > 0.$$

*Proof.* Let $\mathbf{z} \in \Omega \setminus \mathcal{M}$ be arbitrary. Then $f(\mathbf{z}) \neq \mathbf{z}$, so $\mathbf{v}(\mathbf{z}) = f(\mathbf{z}) - \mathbf{z} \neq \mathbf{0}$. By contractiveness and training optimality, $f(\mathbf{z})$ is closer to the fixed-point set $\mathcal{M}$ than $\mathbf{z}$ is. Since $q(\mathbf{z})$ is concentrated on $\mathcal{M}$, we have:

$$q(f(\mathbf{z})) > q(\mathbf{z}) \quad \Rightarrow \quad \log q(f(\mathbf{z})) > \log q(\mathbf{z}).$$

We now Taylor-expand $\log q$ at $\mathbf{z}$ in direction $\mathbf{v}(\mathbf{z})$:

$$\log q(f(\mathbf{z})) = \log q(\mathbf{z}) + \langle \nabla \log q(\mathbf{z}), \mathbf{v}(\mathbf{z}) \rangle + R,$$

where $R = \frac{1}{2}\mathbf{v}^\top(\mathbf{z})\nabla^2 \log q(\xi)\mathbf{v}(\mathbf{z})$, for some $\xi$ between $\mathbf{z}$ and $f(\mathbf{z})$. Since $f$ is contractive, $\|\mathbf{v}(\mathbf{z})\|$ is small and $R = o(\|\mathbf{v}(\mathbf{z})\|)$. Therefore:

$$\langle \nabla \log q(\mathbf{z}), \mathbf{v}(\mathbf{z}) \rangle > -R \quad \Rightarrow \quad \langle \nabla \log q(\mathbf{z}), \mathbf{v}(\mathbf{z}) \rangle > 0.$$

$\square$

**Theorem A.6** (Convergence to latent-space modes). *Let Assumptions 1–4 hold. Let $\mathbf{z}_0 \in \Omega$, and define the iterative sequence:*

$$\mathbf{z}_{t+1} := f(\mathbf{z}_t) = E(D(\mathbf{z}_t)) \quad \text{for all } t \geq 0.$$

*Then:*

1. *The sequence $\{\mathbf{z}_t\}$ converges exponentially fast to a unique fixed point $\mathbf{z}^* \in \mathcal{M}$, satisfying $f(\mathbf{z}^*) = \mathbf{z}^*$.*

2. *At the limit point $\mathbf{z}^*$, we have $\nabla \log q(\mathbf{z}^*) = 0$.*

3. *Moreover, $\mathbf{z}^*$ is a local maximum of the density $q$: the Hessian satisfies*

$$\nabla^2 \log q(\mathbf{z}^*) \prec 0,$$

   *meaning that the Hessian is negative definite.*

*Proof.* **Step 1: Contraction and convergence.** By Assumption 3, $f : \Omega \to \Omega$ is a contraction mapping with contraction constant $L < 1$. Since $\Omega$ is convex and hence complete under $\| \cdot \|$, Banach's fixed-point theorem applies. Therefore:

- There exists a unique fixed point $\mathbf{z}^* \in \Omega$ such that $f(\mathbf{z}^*) = \mathbf{z}^*$.

- For any initial $\mathbf{z}_0 \in \Omega$, the iterates satisfy:

$$\|\mathbf{z}_t - \mathbf{z}^*\| \leq L^t \|\mathbf{z}_0 - \mathbf{z}^*\| \to 0 \quad \text{as } t \to \infty.$$

**Step 2: Stationarity of the limit.** At the fixed point $\mathbf{z}^*$, we have:

$$\mathbf{v}(\mathbf{z}^*) = f(\mathbf{z}^*) - \mathbf{z}^* = \mathbf{0}.$$

Assume for contradiction that $\nabla \log q(\mathbf{z}^*) \neq \mathbf{0}$. By continuity of $\nabla \log q$, there exists a neighborhood $U \ni \mathbf{z}^*$ where $\nabla \log q(\mathbf{z})$ stays close to $\nabla \log q(\mathbf{z}^*)$. Then for nearby $\mathbf{z}$, Lemma A.5 implies:

$$\langle \nabla \log q(\mathbf{z}), \mathbf{v}(\mathbf{z}) \rangle > 0.$$

But continuity of $f$ and the residual $\mathbf{v}(\mathbf{z}) \to \mathbf{0}$ implies the ascent direction vanishes at $\mathbf{z}^*$, which contradicts the assumption that $\nabla \log q(\mathbf{z}^*) \neq \mathbf{0}$. Hence, we conclude:

$$\nabla \log q(\mathbf{z}^*) = \mathbf{0}.$$

**Step 3: Local maximality.** By Lemma A.5 and Assumption 4, the sequence $q(\mathbf{z}_t)$ is strictly increasing and converges to $q(\mathbf{z}^*)$. If $\mathbf{z}^*$ were a saddle point or local minimum, there would exist a direction $\mathbf{u} \in \mathbb{R}^d$ such that the second-order Taylor expansion yields:

$$\frac{d^2}{dt^2} \log q(\mathbf{z}^* + t\mathbf{u}) \bigg|_{t=0} = \mathbf{u}^\top \nabla^2 \log q(\mathbf{z}^*) \mathbf{u} \geq 0,$$

which contradicts the fact that $\mathbf{z}_t \to \mathbf{z}^*$ through ascent directions. Therefore, all second directional derivatives must be negative, implying:

$$\nabla^2 \log q(\mathbf{z}^*) \prec 0.$$

That is, $\mathbf{z}^*$ is a strict local maximum of $q$. $\qquad\square$

## A.2 PROPOSITION 3.1

We report here a detailed formulation of Proposition 3.1 alongside its proof.

**Proposition A.7.** *Let $f = E \circ D : \mathbb{R}^d \to \mathbb{R}^d$ be the composition of an autoencoder's decoder $D$ and encoder $E$, and define the residual vector field $\mathbf{v}(\mathbf{z}) := f(\mathbf{z}) - \mathbf{z}$. Consider the reconstruction loss*

$$L(\mathbf{z}) := \|f(\mathbf{z}) - \mathbf{z}\|^2.$$

*Then, the iteration $\mathbf{z}_{t+1} = f(\mathbf{z}_t)$ corresponds locally to gradient descent on $L$ (i.e., $\mathbf{v}(\mathbf{z}) \propto -\nabla_{\mathbf{z}} L(\mathbf{z})$) if either of the following conditions holds:*

1. *The Jacobian $J_f(\mathbf{z}) \approx \mathrm{Id}$ (i.e., $f$ is locally an isometry).*

2. *$\mathbf{z}$ is near an attractor point $\mathbf{z}^*$ such that $f(\mathbf{z}^*) = \mathbf{z}^*$ and $J_f(\mathbf{z}^*) \approx 0$.*

*Proof.* We begin by expanding the loss:
$$L(\mathbf{z}) = \|f(\mathbf{z}) - \mathbf{z}\|^2 = \langle f(\mathbf{z}) - \mathbf{z}, f(\mathbf{z}) - \mathbf{z}\rangle.$$
Let $\mathbf{v}(\mathbf{z}) := f(\mathbf{z}) - \mathbf{z}$. Then,
$$L(\mathbf{z}) = \|\mathbf{v}(\mathbf{z})\|^2.$$
Now compute the gradient of $L$:
$$\nabla_{\mathbf{z}} L(\mathbf{z}) = \nabla_{\mathbf{z}} \left(\mathbf{v}(\mathbf{z})^\top \mathbf{v}(\mathbf{z})\right)$$
$$= 2 J_{\mathbf{v}}(\mathbf{z})^\top \mathbf{v}(\mathbf{z}),$$
where $J_{\mathbf{v}}(\mathbf{z}) = J_f(\mathbf{z}) - \mathrm{Id}$ is the Jacobian of the residual field.

So we obtain:
$$\nabla_{\mathbf{z}} L(\mathbf{z}) = 2(J_f(\mathbf{z}) - \mathrm{Id})^\top \mathbf{v}(\mathbf{z}).$$
Therefore, if we ask whether $\mathbf{v}(\mathbf{z}) \propto -\nabla_{\mathbf{z}} L(\mathbf{z})$, we need:
$$\mathbf{v}(\mathbf{z}) \propto -(J_f(\mathbf{z}) - \mathrm{Id})^\top \mathbf{v}(\mathbf{z}),$$
which simplifies to:
$$[(J_f(\mathbf{z}) - \mathrm{Id})^\top + \alpha \mathrm{Id}]\mathbf{v}(\mathbf{z}) = 0$$
for some $\alpha > 0$. This holds approximately in the two special cases:

- **Isometry:** If $J_f(\mathbf{z}) \approx \mathrm{Id}$, then $\nabla L(\mathbf{z}) \approx 0$, so $\mathbf{v}(\mathbf{z})$ is nearly stationary—i.e., we are at or near a local minimum.

- **Attractor:** If $\mathbf{z} \approx \mathbf{z}^*$ with $f(\mathbf{z}^*) = \mathbf{z}^*$ and $J_f(\mathbf{z}^*) \approx 0$, then:
$$\nabla L(\mathbf{z}^*) \approx -2\mathbf{v}(\mathbf{z}^*) = 0,$$
and thus $\mathbf{z}^*$ is a fixed point and local minimum of the loss.

In these two cases cases, the dynamics of $\mathbf{z}_{t+1} = f(\mathbf{z}_t)$ follow the direction of steepest descent of $L$, up to scaling. Higher order terms dominate influence the dynamics in the general case.

$\square$

## A.3 ATTRACTORS CONNECTS TO GENERALIZATION ERROR

We report here a detailed formulation of Proposition 3.2 alongside its proof.

**Proposition A.8** (Attractors as prototypes for generalization). *Let $\mathbf{Z}^\star \subset \Omega \subset \mathbb{R}^d$ be a (finite) dictionary of attractors of $f = E \circ D$ in a neighborhood $\Omega$ of latent space, and let $\Pi : \Omega \to \mathbf{Z}^\star$ denote a (measurable) nearest-attractor map $\Pi(\mathbf{z}) \in \arg\min_{\mathbf{u} \in \mathbf{Z}^\star} \|\mathbf{z} - \mathbf{u}\|_2$ (with any fixed tie-breaking rule). If the decoder $D$ is $L_D$-Lipschitz on $\Omega$, then for any test point $\mathbf{x}$ with $\mathbf{z} = E(\mathbf{x}) \in \Omega$:*
$$\left\|\mathbf{x} - F(\mathbf{x})\right\|_2^2 \leq \underbrace{\left\|\mathbf{x} - D(\Pi(\mathbf{z}))\right\|_2^2}_{\text{prototype error}} + \underbrace{L_D^2 \left\|\mathbf{z} - \Pi(\mathbf{z})\right\|_2^2}_{\text{coverage error}}.$$
*Moreover, if $\mathbf{Z}^\star$ is an $\varepsilon$-cover of $\mathrm{supp}\, q$ (the latent marginal), then for all $\mathbf{x}$ with $E(\mathbf{x}) \in \mathrm{supp}\, q$,*
$$\left\|\mathbf{x} - F(\mathbf{x})\right\|_2^2 \leq \left\|\mathbf{x} - D(\Pi(E(\mathbf{x})))\right\|_2^2 + L_D^2 \varepsilon^2,$$
*and the same bound holds in expectation over $\mathbf{x} \sim p_{\text{test}}$ whenever $E(\mathbf{x}) \in \mathrm{supp}\, q$ almost surely.*

We make the following mild assumptions:

(A1) $\mathbf{Z}^* \subset \Omega$ is a (finite) set of attracting fixed points of $f = E \circ D$ contained in an open neighborhood $\Omega \subset \mathbb{R}^d$.

(A2) $D$ is $L_D$-Lipschitz on $\Omega$, i.e., $\|D(\mathbf{z}) - D(\mathbf{u})\| \leq L_D \|\mathbf{z} - \mathbf{u}\|_2$ for all $\mathbf{z}, \mathbf{u} \in \Omega$.

(A3) $\Pi : \Omega \to \mathbf{Z}^\star$ is any measurable selection of nearest points, e.g. $\Pi(\mathbf{z}) \in \arg\min_{\mathbf{u} \in \mathbf{Z}^\star} \|\mathbf{z} - \mathbf{u}\|_2$ .

**Proof of Proposition A.8.** Let $\mathbf{x}$ be any test point with $\mathbf{z} = E(\mathbf{x}) \in \Omega$. Add and subtract $D(\Pi(\mathbf{z}))$:

$$\mathbf{x} - D(E(\mathbf{x})) = \underbrace{\mathbf{x} - D(\Pi(\mathbf{z}))}_{\text{prototype error}} + \underbrace{D(\Pi(\mathbf{z})) - D(E(\mathbf{x}))}_{\text{decoder distortion}}.$$

Taking norms and using Cauchy-Schwarz inequality gives:

$$\|\mathbf{x} - D(E(\mathbf{x}))\|_2^2 \le \|\mathbf{x} - D(\Pi(\mathbf{z}))\|_2^2 + \|D(\Pi(\mathbf{z})) - D(E(\mathbf{x}))\|_2^2.$$

By (A2), $\|D(\Pi(\mathbf{z})) - D(E(\mathbf{x}))\| \le L_D \|\Pi(\mathbf{z}) - \mathbf{z}\|_2$, which yields:

$$\|\mathbf{x} - F(\mathbf{x})\|_2^2 \le \|\mathbf{x} - D(\Pi(\mathbf{z}))\|_2^2 + L_D^2 \|\mathbf{z} - \Pi(\mathbf{z})\|_2^2.$$

This proves the first inequality. If $\mathbf{Z}^\star$ is an $\varepsilon$-cover of $\operatorname{supp} q$, then for any $\mathbf{z} \in \operatorname{supp} q$ we have $\|\mathbf{z} - \Pi(\mathbf{z})\| \le \varepsilon$, hence

$$\|\mathbf{x} - F(\mathbf{x})\|_2^2 \le \|\mathbf{x} - D(\Pi(E(\mathbf{x})))\|_2^2 + L_D^2 \varepsilon^2,$$

for all $\mathbf{x}$ with $E(\mathbf{x}) \in \operatorname{supp} q$. Taking expectations (when $E(\mathbf{x}) \in \operatorname{supp} q$ almost surely) gives the stated bound. □

**Remarks.** (i) The bound decomposes test error into a *prototype term*, $\|\mathbf{x} - D(\Pi(E(\mathbf{x})))\|_2^2$, and a *coverage term*, $L_D^2 \|\mathbf{z} - \Pi(\mathbf{z})\|_2^2$. (ii) Under the assumptions of Theorem 1, $\mathbf{Z}^*$ are attractors with basins; then $\Pi$ aligns with the destination of latent dynamics, tying the algebraic bound to the memorization–generalization picture.

## A.4 ATTRACTORS IN SIMPLE NETWORKS

In this section we characterize attractors in linear and homogeneous networks, showing that in this simple setting is possible to prove converge speed to attractors of the iteration in Eq 3.

**Linear maps.** In the linear case, the encoding map $E$ and decoding $D$ are parametrized by matrices $\mathbf{W}_1 \in \mathbb{R}^{N \times k}$ and $\mathbf{W}_2 \in \mathbb{R}^{k \times N}$ and the only fixed point of the map corresponds to the *origin* if the network is bias free or to a shift of it. The rate of convergence of the iteration in Eq 3 is established by the spectrum of $\mathbf{W}_2^T \mathbf{W}_1$, and the iteration is equivalent to shrinking the input in the direction of the eigenvectors with associated eigenvalue $\lambda < 1$. In case the eigenvectors of the trained AEs are aligned with the optimal solution given by the Principal Component Analysis (PCA) decomposition of the data $\mathbf{X} = \boldsymbol{\Phi}\boldsymbol{\Lambda}\boldsymbol{\Phi}^*$, then the latent vector field vanished as the mapping is isometric and no contraction occurs.

**Homogeneous maps.** In the case of homogeneous neural networks, the network satisfies $F(c\mathbf{x}) = c^\alpha F(\mathbf{x})$ with $c \in \mathbb{R}$ for some $\alpha$. For example, this holds for ReLU networks without biases with $\alpha = 1$, which learn a piecewise linear mapping. The input-output mapping can be rewritten as:

$$F(\mathbf{x}) = J_F(\mathbf{x})\mathbf{x} \tag{6}$$

A similar observation was made in for denoising networks in (Kadkhodaie et al., 2023), we remark here its generality.

This equality implies that we can rewrite Equation 3 as :

$$\mathbf{z}_{t+1} = J_f(\mathbf{z}_t)\mathbf{z}_t = \sum_i \lambda_i \phi_i \mathbf{z}_t \tag{7}$$

Where $\sum_i \lambda_i \phi_i$ is the eigendecomposition of $J_f(\mathbf{z})$. Since in the proximity of an attractor $max\lambda(J_F(\mathbf{z}^*)) <= 1$, the iterations shrink directions corresponding to the eigenvectors of the Jacobian by the corresponding eigenvalue. This allows us to derive the following result on the speed of convergence.

**Proposition A.9** (informal, proof in Appendix A.5). *The error $e_t = \|\mathbf{z}_t - \mathbf{z}^*\|$ converge exponentially to an $\epsilon$ depending on the spectral norm $\|J_f(\mathbf{z}^*)\|_\sigma$, according to the formula $\frac{\log(\frac{\epsilon}{\|\mathbf{e}_0\|})}{\log(\|J_f(\mathbf{z}^*)\|_\sigma)}$ that provides an estimate for the number of iterations $T$ to converge to the attractor.*

In Figure 11 in the Appendix, we measure how well the convergence formula predicts the measured number of iterations to converge to an attractor, on a fully non linear AE with biases, where the assumptions of the current section are less likely to hold. The error in the estimate will be higher, when the initial condition $\mathbf{z}_0$ is far from the attractor, as higher order terms dominate the dynamics, and the first order Taylor approximation accumulates more error.

## A.5 Proof of Proposition A.9

We report here a detailed formulation of Proposition A.9 alongside its proof.

**Proposition A.10.** *Let $f : \mathbb{R}^n \to \mathbb{R}^n$ be a differentiable function with a fixed point $\mathbf{z}^* \in \mathbb{R}^n$ such that $f(\mathbf{z}^*) = \mathbf{z}^*$. Assume the Jacobian $J_f(\mathbf{z}^*)$ has spectral norm $\|J_f(\mathbf{z}^*)\|_\sigma = \rho < 1$. Then, for initial point $\mathbf{z}_0$ sufficiently close to $\mathbf{z}^*$, the sequence $\{\mathbf{z}_t\}$ defined by the iteration $\mathbf{z}_{t+1} = f(\mathbf{z}_t)$ satisfies:*

$$\|\mathbf{z}_t - \mathbf{z}^*\| \leq \|\mathbf{e}_0\| \cdot \rho^t,$$

*and the number of iterations $T$ needed to reach an error $\|\mathbf{z}_T - \mathbf{z}^*\| \leq \epsilon$ is bounded by:*

$$T \geq \frac{\log(\epsilon/\|\mathbf{e}_0\|)}{\log(\rho)}.$$

*Proof.* Let $\mathbf{e}_t = \mathbf{z}_t - \mathbf{z}^*$ denote the error at iteration $t$. We are given that $f$ is differentiable around $\mathbf{z}^*$, and that $f(\mathbf{z}^*) = \mathbf{z}^*$. Applying the first-order Taylor expansion of $f$ around $\mathbf{z}^*$, we obtain:

$$f(\mathbf{z}_t) = f(\mathbf{z}^*) + J_f(\mathbf{z}^*)(\mathbf{z}_t - \mathbf{z}^*) + R(\mathbf{z}_t),$$

where $R(\mathbf{z}_t)$ is the Taylor remainder satisfying $\|R(\mathbf{z}_t)\| = o(\|\mathbf{z}_t - \mathbf{z}^*\|)$. Since $f(\mathbf{z}^*) = \mathbf{z}^*$, this becomes:

$$\mathbf{z}_{t+1} = f(\mathbf{z}_t) = \mathbf{z}^* + J_f(\mathbf{z}^*)(\mathbf{z}_t - \mathbf{z}^*) + R(\mathbf{z}_t).$$

Thus, the error evolves as:

$$\mathbf{e}_{t+1} = \mathbf{z}_{t+1} - \mathbf{z}^* = J_f(\mathbf{z}^*)\mathbf{e}_t + R(\mathbf{z}_t).$$

For $\mathbf{z}_t$ sufficiently close to $\mathbf{z}^*$, we can neglect the higher-order term $R(\mathbf{z}_t)$ in comparison to the leading linear term. Hence, the error evolves approximately as:

$$\|\mathbf{e}_{t+1}\| \leq \|J_f(\mathbf{z}^*)\|_\sigma \cdot \|\mathbf{e}_t\| + o(\|\mathbf{e}_t\|).$$

By continuity of $f$, there exists a neighborhood $\mathcal{U}$ of $\mathbf{z}^*$ where $\|J_f(\mathbf{z})\|_\sigma \leq \rho + \delta < 1$ for some $\delta > 0$. Choosing $\mathbf{z}_0 \in \mathcal{U}$ and letting $R(\mathbf{z}_t) = 0$ (first-order approximation), the error satisfies:

$$\|\mathbf{e}_t\| \leq \|\mathbf{e}_0\| \cdot \rho^t.$$

We now ask: for which $T$ does $\|\mathbf{e}_T\| \leq \epsilon$? We solve:

$$\|\mathbf{e}_0\| \cdot \rho^T \leq \epsilon.$$

Dividing both sides by $\|\mathbf{e}_0\|$ and taking logarithms yields:

$$\rho^T \leq \frac{\epsilon}{\|\mathbf{e}_0\|} \quad \Rightarrow \quad T \cdot \log(\rho) \leq \log\left(\frac{\epsilon}{\|\mathbf{e}_0\|}\right).$$

Since $\log(\rho) < 0$, we reverse the inequality when dividing:

$$T \geq \frac{\log(\epsilon/\|\mathbf{e}_0\|)}{\log(\rho)}.$$

This proves that the error converges exponentially and that the number of iterations required to reach error $\epsilon$ is lower bounded by the expression above. $\square$

## B Additional experiments

### B.1 Bias at initialization

As stated in the method section of the main paper, To empirically measure to what extent model are contractive at initialization we sample $N = 1000$ samples from $\mathcal{N}(0, I)$, and we map them trough 12 different vision backbones, initialized randomly across 10 seeds. We measure the ratio between the input variance $\sigma_{in} = 1$ and output variance $\sigma_{out}$ and we report in Figure 6 .

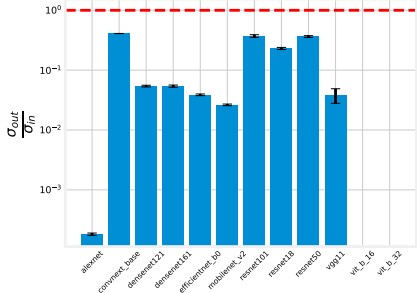

Figure 6: *Contraction at initialization* Variance preserving ratio at initialization of torchvision models: all models considered have a ratio $< 1$, indicating that the map at initialization is contractive.

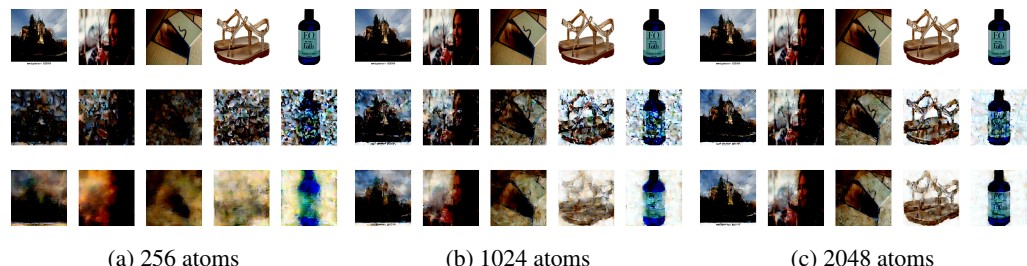

|  (a) 256 atoms | (b) 1024 atoms | (c) 2048 atoms |

Figure 7: Visualization of reconstructions from the data-free sample recovery experiment of Figure 4: Visualizing on `Laion2B` reconstructions of five random samples. First row: input samples; second row: reconstructions from orthogonal basis; third row reconstructions from attractors of noise.

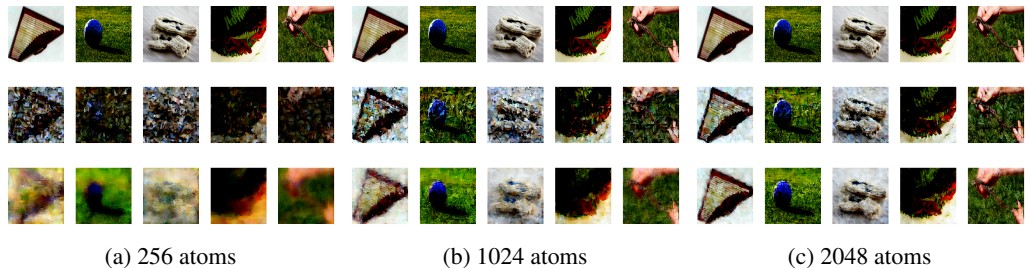

|  (a) 256 atoms | (b) 1024 atoms | (c) 2048 atoms |

Figure 8: Visualization of reconstructions from the data-free sample recovery experiment of Figure 4: Visualizing on `Imagenet1k` reconstructions of five random samples. First row: input samples; second row: reconstructions from orthogonal basis; third row reconstructions from attractors of noise.

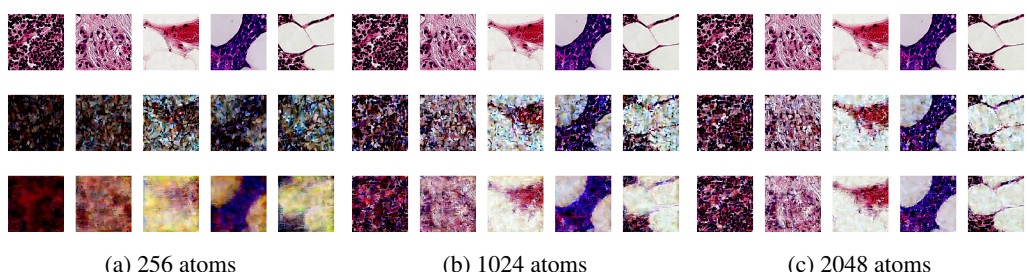

|  (a) 256 atoms | (b) 1024 atoms | (c) 2048 atoms |

Figure 9: Visualization of reconstructions from the data-free sample recovery experiment of Figure 4: Visualizing on `Camelyon17` reconstructions of five random samples. First row: input samples; second row: reconstructions from orthogonal basis; third row reconstructions from attractors of noise.

### B.2    QUALITATIVE RESULTS: DATA-FREE WEIGHT PROBING

We report additional qualitative results from the data free weight probing experiment in section 4.1, in Figures 7,8,9, observing that the reconstructions from attractors as a functions of the number of atoms used, are superior with respect to the orthogonal basis baseline.

### B.3    ADDITIONAL RESULTS ON LATENT TRAJECTORIES

In Table 1 we extend results of the experiment in Fig. 5, adding two additional baselines as OOD detection score: Mahanabolis distance to the on in-distribution features and reconstruction error. In both cases statistics using the latent trajectories are more discriminative than using scores based on features.

|  | SUN397 | | Places365 | | Texture | | iNaturalist | |
|---|---|---|---|---|---|---|---|---|
| Method | FPR ↓ | AUROC ↑ | FPR ↓ | AUROC ↑ | FPR ↓ | AUROC ↑ | FPR ↓ | AUROC ↑ |
| KNN | 100.00 | 42.60 | 100.00 | 32.40 | 34.50 | 89.40 | 86.40 | 68.60 |
| Mahalanobis | 82.00 | 58.00 | 88.00 | 45.00 | 43.00 | 90.00 | 31.00 | 87.00 |
| Reconstruction | 91.80 | 49.50 | 91.60 | 50.90 | 98.40 | 49.10 | 99.20 | 23.60 |
| d(Attractors) | **29.60** | **91.20** | **29.90** | **91.00** | **25.90** | **92.60** | **29.90** | **91.30** |

Table 1: *FPR and AUROC scores extended*: results of the experiment in Figure 5 extended, adding reconstruction error baseline and Mahalanobis distance on the features space.

#### B.3.1    HISTOGRAMS OOD DETECTION

We report in Figure 10 additional histograms of In-Distribution vs Out-Of-Distribution scores, corresponding to the experiment in Section 4.2.

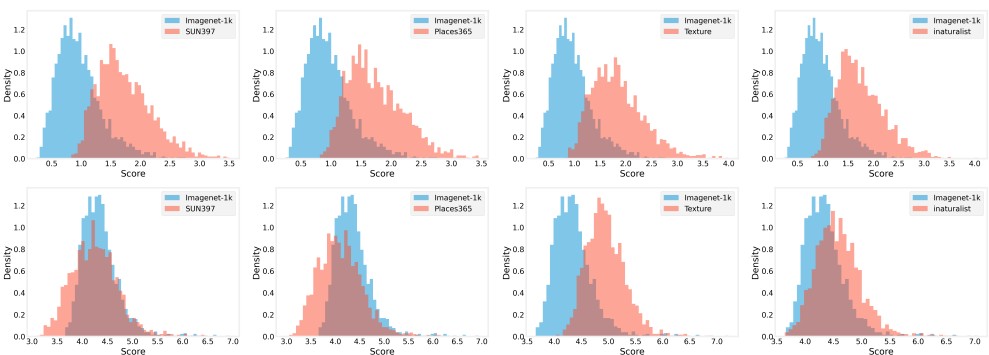

Figure 10: Histograms from OOD detections

### B.4    ANALYSIS SPEED OF CONVERGENCE

In figure 11 we report an empirical validation of proposition A.9, for a 2 dimensional bottleneck convolution autoencoder trained on MNIST, by sampling points in the training set and measuring their convergence speed in terms of number of iteration. We remark that assumptions for the proposition don't hold for this model, as the farer an initial condition is from an attractor the more higher order terms will dominate in the dynamics, making the evaluation of the spectrum of the Jacobian at the attractor insufficient. We observe nevertheless, that the convergence speed bounds still correlates, providing a looser estimate for the number of iteration needed to converge towards an attractors with error $\epsilon$.

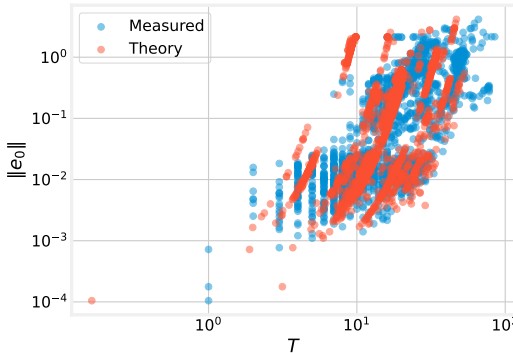

Figure 11: *Converge speed analysis* We plot the convergence speed in terms of number of iterations vs the error (distance to the attractors) of the initial condition on a convolutional autoencoders trained on MNIST. The estimated convergence has a Pearson correlation coefficient of 0.67 to the true number of iterations to converge. Higher error in the estimates occur for farer initial conditions.

### B.4.1 CONVERGENCE RATES IN PRETRAINED MODELS

In Figure 12 we plot the rate of convergence in terms of latent residuals norms of the latent trajectories in ViT MAE autoencoders from 2000 samples drawn from the `Imagenet` dataset. We plot (a) the latent residual norm as function of the number od the iteration, showing that it approaches zero exponentially fast; (b) the norm of the embedding at each iteration. showing that they are bounded and the trajectory don't diverge.

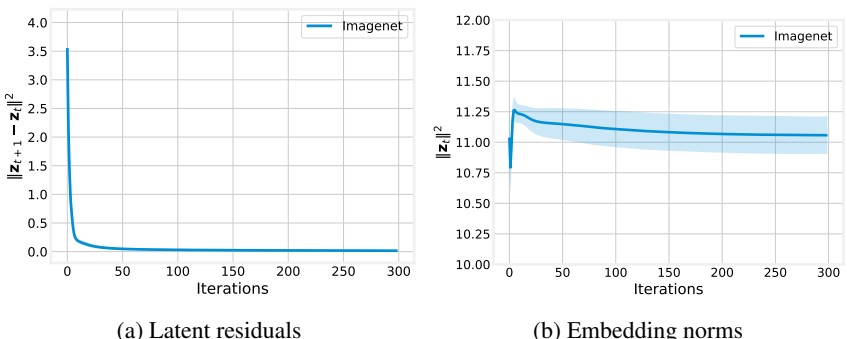

(a) Latent residuals           (b) Embedding norms

Figure 12: *Latent trajectories in masked autoencoders are contractive.*: we plot norms of the latent residuals across iterations in a ViTMAE model (a) and norms of embeddings across iterations (b).

### B.5 RANK OF ATTRACTOR MATRIX

In Figure 13 we report an alternative measure of generalization of the attractors computed in the experiment in Figure 2 and in Figure 3. Specifically, we consider the matrix $\mathbf{X}^* = D(\mathbf{Z})^*$, and we compute its singular value decomposition $\mathbf{U}^* diag(\mathbf{s}^*)\mathbf{V}^* = \mathbf{X}^*$. We define the generalization entropy as the number of eigenvectors needed to explain $90\%$ of the variance of the matrix. We report this measure for both the models employed in the experiment in Figure2 and in Figure 3, showing that attractors are more expressive as the model generalized better, as well during training.

### B.6 MEMORIZATION OVERFITTING REGIME

In this section, we test the memorization capabilities of models in different (strong) overparametrization regimes, similarly as tested in (Kadkhodaie et al., 2023) for diffusion models and in (Radhakrishnan et al., 2020; Zhang et al., 2019) for the extremely overparametrized case (network trained on few samples). We remark that this case corresponds to an overfitting regime of the network, as opposed

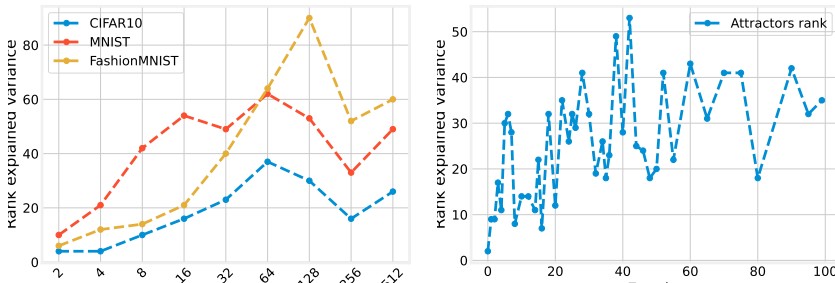

Figure 13: *Generalization increase ranks of attractors matrix*: entropy rank as a function of the bottleneck dimension (*left*) and during training, as a function of the number of epochs (*right*). Transitioning from memorization to generalization

as underfitting (over-regularized) regime showed in the main paper. We train a convolutional autoencoder on subsamples of the CIFAR, MNIST, FashionMNIST datasets, with bottleneck dimension $k = 128$ and weight decay $1e - 4$. In Figure 14 we plot the memorization coefficient as a function of the dataset size, showing that networks trained on less data( strong overparametrization) tend to memorize more the training data.

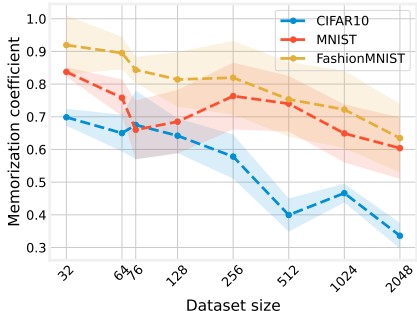

Figure 14: *Strong overparametrization favors memorization*: Memorization coefficient as a function of the dataset size on CIFAR, MNIST, FashionMNIST, showing that networks trained on less data (*strongly overparametrized*) tend to memorize more the training data.

### B.7 DATA-FREE WEIGHT PROBING: ADDITIONAL RESULTS

We replicate the experiment in 4.1, on the larger XL version of the Stable diffusion architecture (Rombach et al., 2022), observing that attractors from noise still form an informative dictionary of signals which scales better w.r.t, to a random orthogonal basis.

### B.8 BEYOND AUTOENCODER MODELS

In this section we preliminary explore the extent to which latent vector exist beyond pretrained autoencoder models.

### B.8.1 LATENT VECTOR FIELDS IN SELF SUPERVISED MODELS

We first consider the case discriminative self-supervised (SSL) models such as DINOv2 Oquab et al. (2023) and SigLIP2 Tschannen et al. (2025). A straightforward idea to extend our framework to encoder-only models, is the one of training a decoder on top of the frozen encoder model. To this end we employ the decoders trained in Zheng et al. (2025), which provide pretrained decoders models for DINOv2 and SigLIP2.

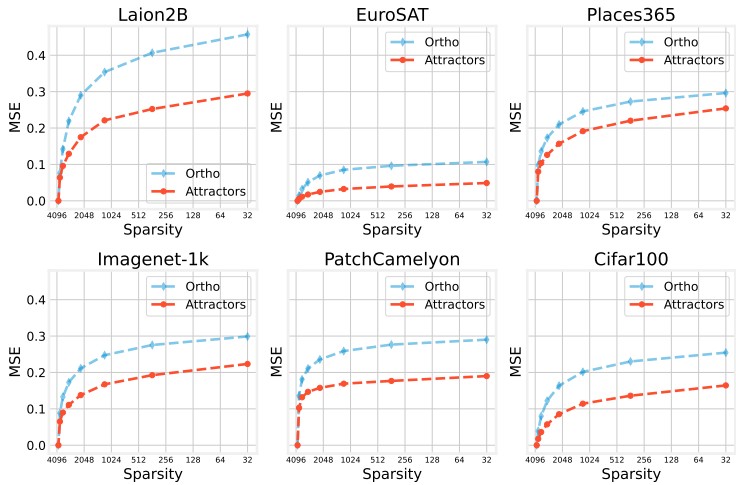

Figure 15: *Data-free weight probing Stable diffusion XL AE* : the results on the larger version of the table diffusion autoencoder confirm that attractors from noise form an informative dictionary of signals to reconstruct different datasets.

To assess how the latent vector field behaves, we sample 500 samples from the `Imagenet1k` training dataset, and compute 300 iterations of Eq 3. We monitor (a) the norm of the residuals in the output space (i.e. reconstruction error with respect to previous iteration) (b) the norm of the residuals in the latent space (i.e. the elements vector field at current time step) (c) the average norm of the embeddings at every step. We show the results in Figure 16 observe that the reconstruction residuals and the latent space residuals both monotonically decrease at the beginning, with the former approaching 0, demonstrating a contraction behavior and then stabilize. This together with the observation that the norm of the embeddings is bounded and does not increase across iterations makes us conclude that the latent vector field exists, does not diverge, and is well behaved. The latent space residuals stabilizing around iteration 150 highlight that the eigenvalues of $J_f(z_t)$ are approaching 1, indicating either that the iteration has reached an attractor region (as opposed to a point) or that the the map is still contractive but with a constant very close to 1.

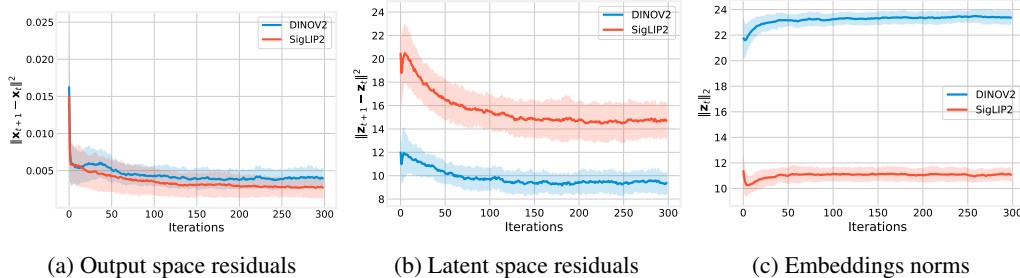

(a) Output space residuals      (b) Latent space residuals      (c) Embeddings norms

Figure 16: *SSL models induce well-behaved latent vector fields*: For latent space iterations in DINOv2 and SigLIP2, we plot norm of residuals in output space (a);latent residuals norms (b); and norms of embeddings across iterations (c).

### B.8.2    LATENT VECTORS FIELDS IN LLMS

We now consider the case of next token predictors models, in particular case light weight large language models. In this case we consider the input-output mapping in representation space, by iterating representation computed after the embedding layer to the final layer before the softmax, therefore $f$ corresponds to the entire residual stream of the model. As models we consider Qwen3-0.6 B Yang et al. (2025) and Smol-LM2 Allal et al. (2025). To test whether the latent vectors field are we sample 1000 sample from the `Pile` dataset Gao et al. (2020) and iterate Eq. 3 for 300 iteration

monitoring residuals in the latent space (final layer) (a) and the norm of the final layer embeddings (b). In Figure 17 we plot the results, observing that both models show contractive behavior (a) ithout diverging, given the bounded norms (b).

Taken together these results give us preliminary evidence that latent vector fields exist and are well defined beyond the case of autoencoder model, but we stress that further analysis should be performed in order to study them, and analyze their attractor modes whenever they exists. Also the impact of different architectural elements, for example for LLMs the role of positional embeddings, (e.g. rotary position embeddings Su et al. (2024) as opposed to absolute ones) pose questions on the resulting dynamics. We believe that studying properties of the latent vector field induced by models beyond autoencoders hold promise to understand mechanistically the properties of the models, and we find promising to study recent phenomenon such as latent space reasoning Zhu et al. (2025).

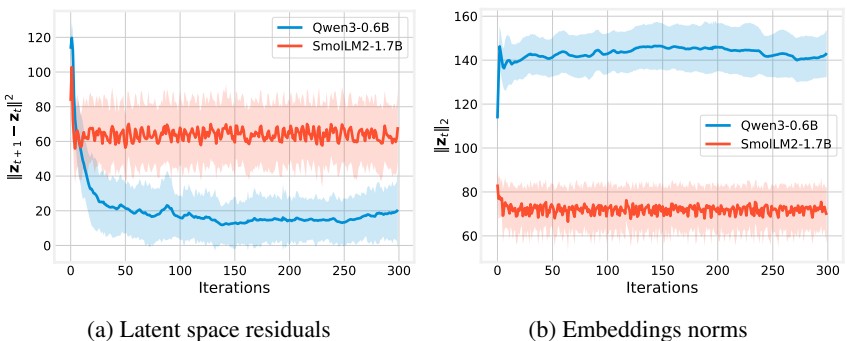

(a) Latent space residuals   (b) Embeddings norms

Figure 17: *LLM models can induce well-behaved latent vector fields*: For latent space iterations in Qwen3-0.6B and SmolLM2-1.7b, we plot the latent residuals norms(a); and norms of embeddings across iterations (b).

### B.9 QUALITATIVE VISUALIZATION OF ATTRACTORS

In this section we provide more visualizations from decoded attractors other than the ones in Figure 2 (c). In Figure 18 we report examples of decoded attractors from the latent space of the Stable Diffusion autoencoder. Attractors don't show semantically meaningful patterns but rather geometrical and texture patterns,resembling elements of a basis, similarly to the last rows of Figure 2 (c).

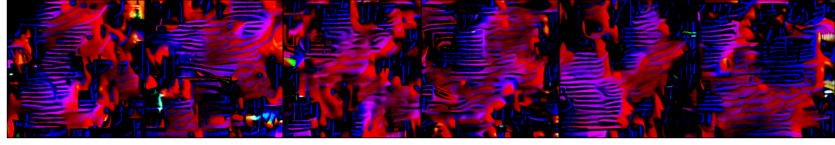

Figure 18: *Visualization of decoded attractors from Stable Diffusion AE.*

Table 2: Architecture of the AutoencoderKL in Stable Diffusion

| Layer | Details | Output Shape |
|---|---|---|
| **Encoder** | | |
| Input | $3 \times 256 \times 256$ image | $3 \times 256 \times 256$ |
| Conv2D | $3 \rightarrow 128$, $3 \times 3$, stride 1 | $128 \times 256 \times 256$ |
| ResNet Block | 128 channels | $128 \times 256 \times 256$ |
| Conv2D | $128 \rightarrow 256$, $3 \times 3$, stride 2 | $256 \times 128 \times 128$ |
| ResNet Block | 256 channels | $256 \times 128 \times 128$ |
| Conv2D | $256 \rightarrow 512$, $3 \times 3$, stride 2 | $512 \times 64 \times 64$ |
| ResNet Block | 512 channels | $512 \times 64 \times 64$ |
| Conv2D | $512 \rightarrow 512$, $3 \times 3$, stride 2 | $512 \times 32 \times 32$ |
| ResNet Block | 512 channels | $512 \times 32 \times 32$ |
| Conv2D | $512 \rightarrow 512$, $3 \times 3$, stride 2 | $512 \times 16 \times 16$ |
| ResNet Block | 512 channels | $512 \times 16 \times 16$ |
| Conv2D | $512 \rightarrow 512$, $3 \times 3$, stride 2 | $512 \times 8 \times 8$ |
| ResNet Block | 512 channels | $512 \times 8 \times 8$ |
| Mean and Log Variance | $512 \rightarrow 4$, $1 \times 1$ | $4 \times 8 \times 8$ |
| Sampling | Reparameterization trick | $4 \times 8 \times 8$ |
| **Decoder** | | |
| Conv2D | $4 \rightarrow 512$, $3 \times 3$, stride 1 | $512 \times 8 \times 8$ |
| ResNet Block | 512 channels | $512 \times 8 \times 8$ |
| Upsample | Scale factor 2 | $512 \times 16 \times 16$ |
| ResNet Block | 512 channels | $512 \times 16 \times 16$ |
| Upsample | Scale factor 2 | $512 \times 32 \times 32$ |
| ResNet Block | 512 channels | $512 \times 32 \times 32$ |
| Upsample | Scale factor 2 | $512 \times 64 \times 64$ |
| ResNet Block | 512 channels | $512 \times 64 \times 64$ |
| Upsample | Scale factor 2 | $256 \times 128 \times 128$ |
| ResNet Block | 256 channels | $256 \times 128 \times 128$ |
| Upsample | Scale factor 2 | $128 \times 256 \times 256$ |
| ResNet Block | 128 channels | $128 \times 256 \times 256$ |
| Conv2D | $128 \rightarrow 3$, $3 \times 3$, stride 1 | $3 \times 256 \times 256$ |
| **Output: Reconstructed Image** | | |

## C  REGULARIZED AUTOENCODER ENFORCE CONTRACTIVENESS

In Table 3, we list many AE variants including denoising AEs (DAEs) (Vincent et al., 2008), sparse AEs (SAEs) (Ng et al., 2011), variational AEs (VAEs) (Kingma et al., 2013) and other variants (Rifai et al., 2011; Alain and Bengio, 2014; Gao et al., 2024) and show how their objectives enforce local contractive solutions around training points.

We provide below a precise connection for the case of weight decay and feed-forward networks.

**Example: weight decay**  Consider an $L$-layer feed-forward autoencoder $F_\theta = D_\theta \circ E_\theta$, where each layer is defined as $h_\ell = \phi_\ell(W_\ell h_{\ell-1} + b_\ell)$ and the activations $\phi_\ell$ are 1-Lipschitz (e.g., ReLU, GELU, SiLU, *tanh*). The overall map can be written as

$$F_\theta(x) = W_L \phi_{L-1}\big(W_{L-1} \cdots \phi_1(W_1 x)\big).$$

Let $D_\ell(x) = \mathrm{diag}(\phi'_\ell(W_\ell h_{\ell-1}(x)))$ be the diagonal Jacobian of the activation at layer $\ell$. By the chain rule,

$$J_\theta(x) = W_L D_{L-1}(x) W_{L-1} \cdots D_1(x) W_1.$$

Since each activation is 1-Lipschitz, $\|D_\ell(x)\|_2 \leq 1$ for all $\ell$, and therefore

$$\|J_\theta(x)\|_2 \leq \prod_{\ell=1}^{L} \|W_\ell\|_2.$$

A sufficient condition for local contraction is thus

$$\prod_{\ell=1}^{L} \|W_\ell\|_2 < 1 \quad \Rightarrow \quad \|J_\theta(x)\|_2 < 1.$$

Training with weight decay minimizes $\mathcal{L}(\theta) + \lambda \sum_{\ell=1}^{L} \|W_\ell\|_F^2$, and since $\|W_\ell\|_2 \leq \|W_\ell\|_F$, this suppresses the product of spectral norms appearing above:

$$\prod_{\ell=1}^{L} \|W_\ell\|_2 \leq \prod_{\ell=1}^{L} \|W_\ell\|_F,$$

biasing the solution toward $\|J_\theta(x)\|_2 < 1$ even without an explicit Jacobian penalty.

| Autoencoder | Regularizer $\mathcal{R}_\theta(\mathbf{x})$ | Description | Effect on Contractiveness |
|---|---|---|---|
| **Standard AE (rank $\leq k$)** | – (bottleneck dimension $k$) | No explicit regularization, but dimensionality constraint induces compression | Bottleneck limits Jacobian rank: $\mathrm{rank}(J_{f_\theta}(\mathbf{x})) \leq k$ |
| **Plain AE with Weight Decay** | $\|\mathbf{W}_E\|_F^2 + \|\mathbf{W}_D\|_F^2$ | L2 penalty on encoder and decoder weights | Reduces $\|J_{f_\theta}(\mathbf{x})\| \leq L_\sigma^2 \|\mathbf{W}_D\|\|\mathbf{W}_E\|$ by shrinking weight norms |
| **Deep AE with Weight Decay** | $\sum_\ell \|\mathbf{W}_E^{(\ell)}\|_F^2 + \|\mathbf{W}_D^{(\ell)}\|_F^2$ | Weight decay across multiple layers of deep encoder/decoder | Layerwise shrinkage enforces smoother, more contractive composition $J_{f_\theta}(\mathbf{x})$ |
| **R-Contractive AE** (Alain and Olivier, 2013) | $\|J_{f_\theta}(\mathbf{x})\|_F^2$ | Penalizes Jacobian of the full map $f_\theta = D_\theta \circ E_\theta$ | Encourages stability of reconstruction: $f_\theta(\mathbf{x}) \approx f_\theta(\mathbf{x} + \delta)$ |
| **Contractive AE** (Rifai et al., 2011) | $\|J_{E_\theta}(\mathbf{x})\|_F^2 = \sum_{i=1}^k \|J_{E_i}(\mathbf{x})\|^2$ | Penalizes encoder Jacobian norm | Explicitly enforces local flatness of encoder |
| **Sparse AE (KL, sigmoid)** (Ng et al., 2011) | $\sum_{i=1}^k \mathrm{KL}(\rho \,\|\, \hat{\rho}_i), \quad \hat{\rho}_i = \mathbb{E}_\mathbf{x}[E_i(\mathbf{x})]$ | Enforces low average activation under sigmoid | Saturated units $\Rightarrow J_{E_i}(\mathbf{x}) \approx 0 \Rightarrow \|J_{E_\theta}(\mathbf{x})\|$ small |
| **Sparse AE (L1, ReLU)** (Ng et al., 2011) | $\|E_\theta(\mathbf{x})\|_1$ | Promotes sparsity of ReLU activations | Inactive ReLU units have zero derivatives $\Rightarrow$ sparse Jacobian $J_{E_\theta}(\mathbf{x})$ |
| **Denoising AE** (Vincent et al., 2008; Alain and Bengio, 2014) | $\mathbb{E}_{\tilde{\mathbf{x}}\sim q(\tilde{\mathbf{x}}|\mathbf{x})}\|\mathbf{x} - f_\theta(\tilde{\mathbf{x}})\|^2$ | Reconstructs from noisy input | For small noise: $f_\theta(\mathbf{x}) - \mathbf{x} \approx \sigma^2 \nabla \log p(\mathbf{x})$, a contractive vector field $J_{f_\theta}(\mathbf{x})$ |
| **Variational AE (VAE)** (Kingma et al., 2013) | $\mathrm{KL}(q_\theta(\mathbf{z}|\mathbf{x}) \,\|\, p(\mathbf{z}))$, where $q(\mathbf{z}|\mathbf{x}) = \mathcal{N}(\mu(\mathbf{x}), \Sigma(\mathbf{x}))$ | Encourages latent distribution to match prior | Smooths $\mu(\mathbf{x}), \Sigma(\mathbf{x})$; penalizes sharp variations in encoder $J_\mu(\mathbf{x}), J_\Sigma(\mathbf{x})$ |
| **Masked AE (MAE)** (He et al., 2022) | $\mathbb{E}_M \left[ \|M \odot (\mathbf{x} - f_\theta(M \odot \mathbf{x}))\|^2 \right]$ | Learns to reconstruct from partial input | Promotes invariance to missing entries $\Rightarrow \|J_{f_\theta}(\mathbf{x})\|$ low |

Table 3: Unified formulation of autoencoder variants as minimizing reconstruction error plus a regularizer that encourages local contractiveness.

## D   ADDITIONAL IMPLEMENTATION DETAILS

Table 4: Architecture of the autoencoder employed in experiments on `CIFAR`, `MNIST`, `FashionMNIST` datasets

| Layer | Details | Output Shape |
|---|---|---|
| **Encoder** | | |
| Input | $C \times H \times W$ image | $C \times H \times W$ |
| Conv2D | $C \to d$, $3 \times 3$, stride 2, pad 1 | $d \times \frac{H}{2} \times \frac{W}{2}$ |
| Conv2D | $d \to 2d$, $3 \times 3$, stride 2, pad 1 | $2d \times \frac{H}{4} \times \frac{W}{4}$ |
| Conv2D | $2d \to 4d$, $3 \times 3$, stride 2, pad 1 | $4d \times \frac{H}{8} \times \frac{W}{8}$ |
| Conv2D | $4d \to 8d$, $3 \times 3$, stride 2, pad 1 | $8d \times \frac{H}{16} \times \frac{W}{16}$ |
| Flatten | — | $8d \cdot \frac{H}{16} \cdot \frac{W}{16}$ |
| Bottleneck | Linear layer or projection | $z$ (latent code) |
| **Decoder** | | |
| Unflatten | $z \to 8d \times \frac{H}{16} \times \frac{W}{16}$ | $8d \times \frac{H}{16} \times \frac{W}{16}$ |
| ConvTranspose2D | $8d \to 4d$, $3 \times 3$, stride 2, pad 1, output pad 1 | $4d \times \frac{H}{8} \times \frac{W}{8}$ |
| ConvTranspose2D | $4d \to 2d$, $3 \times 3$, stride 2, pad 1, output pad 1 | $2d \times \frac{H}{4} \times \frac{W}{4}$ |
| ConvTranspose2D | $2d \to d$, $3 \times 3$, stride 2, pad 1, output pad 1 | $d \times \frac{H}{2} \times \frac{W}{2}$ |
| ConvTranspose2D | $d \to C$, $3 \times 3$, stride 2, pad 1, output pad 1 | $C \times H \times W$ |
| **Output: Reconstructed Image** | | |

For the experiments in Figures 2, 3, 14 we considered convolutional autoencoders with architecture specified in Table 4. We train the models with Adam (Kingma, 2014) with learning rate $5e - 4$, linear step learning rate scheduler, and weight decay $1e - 4$ for 500 epochs. For the experiements in section 4.1, we use the pretrained autoencoder of (*Rombachet al.*, 2022). We report in Table 2 the architectural details and we refer to the paper for training details. For the experiments in section 4.2, we use the pretrained autoencoder of (*Heet al.*, 2022), considering the base model. We refer to the paper for training details. For computing attractors in experiments in Figures 2, 3, 14, we compute the iterations $\mathbf{z}_{t+1} = f(\mathbf{z}_t)$ until $\|f(\mathbf{z}_{t+1}) - f(\mathbf{z}_t)\|_2^2 < 1e - 6$ or reaching $t = 3000$. Similarly in experiments in Sections 4.1, 4.2, we compute the iterations $\mathbf{z}_{t+1} = f(\mathbf{z}_t)$ until $\|f(\mathbf{z}_{t+1}) - f(\mathbf{z}_t)\|_2^2 < 1e - 5$ or reaching $t = 500$. All experiments are performed on a GPU NVIDIA 3080TI in Python code, using the PyTorch library (Paszke et al., 2019).

