# OpenReview forum: "Navigating the Latent Space Dynamics of Neural Models"
_ICLR.cc/2026/Conference — ICLR 2026 Oral_

### Official Review · Reviewer_XuAT · 2025-10-19

**Soundness:** 3
**Presentation:** 3
**Contribution:** 3
**Rating:** 6
**Confidence:** 2

**Summary:**

This paper proposes viewing autoencoder models as latent dynamical systems, where iterating the mapping $f=E\circ D$ defines a latent vector field and reveals attractors that capture the model’s memorization and generalization behavior.

The authors connect the local contractivity of this mapping to the emergence of attractors and use them for practical analyses such as (1) distinguishing memorization vs. generalization regimes, and (2) performing data-free probing and out-of-distribution (OOD) detection by analyzing trajectories toward these attractors.

In addition to theoretical connections and remarks, empirical results are shown using autoencoders, a diffusion-model autoencoder, and a large-scale vision model utilizing masked autoencoders.

**Strengths:**

- The idea of treating $E\circ D$ as a dynamical system in latent space is novel and intuitive, providing a unifying perspective on autoencoders.
- The framework is validated across various architectures, including autoencoders, a pretrained diffusion AE, and a large-scale vision model, showing that attractors can indeed be identified even in large-scale, complex models.
- Using attractors derived from noise, without requiring access to source training data, to reconstruct meaningful images is an interesting way to explore what information the model stores in its weights, opening up interpretability and compression directions.
- The paper provides clear and well-motivated definitions of concepts such as contractivity and attractors, and links them intuitively to properties of the model’s Jacobian, which helps make the overall framework more interpretable and understandable.

**Weaknesses:**

- As the authors acknowledged in the discussion, it remains uncertain whether the approach applies to widely used forecasting or next-token prediction models, or encoder-only architectures. Since such models dominate modern representation learning, discussing whether attractors exist or can be defined meaningfully in these settings would strengthen the impact. To examine this, would training a lightweight decoder on top of a frozen encoder (without modifying the encoder weights) help reveal similar attractor dynamics?
This could clarify whether the attractor framework extends beyond autencoder-based models.
- While the authors show that attractors inferred from noise can reconstruct the actual inputs, it is not entirely clear whether these attractors correspond to actual training examples, or how one can infer generalization capacity without explicitly comparing attractors to the training data.
- The theoretical analysis assumes local contractivity, which potentially does not hold globally. Empirically, as long as stable attractors can be identified, the proposed approach appears to remain valid. Nevertheless, it would be good to quantify how much of the latent space exhibits convergent dynamics, characterize the stability of these attractors, and report how long or how many iterations are typically required to discover them beyond the MNIST example.
- Is the KNN analysis performed using latent embeddings of training data or the attractors identified by training data? How sensitive are the KNN results in Figure 5a to the choice of the number of neighbors K? Would similar results hold for smaller values of K or with adaptive neighborhood sizes? Also, for the proposed attractor trajectory-based scoring, is the distance computed with the nearest training attractor's trajectory or averaged across all training attractors?
- Following up on the previous item, how does the proposed attractor trajectory-based scoring compare with other standard OOD detection metrics, such as the Mahalanobis distance in latent space or the reconstruction loss (MSE) in input space?
- Similarly, would OOD detection performance remain similar if attractors were computed from Gaussian noise?
- Please provide the definition of FPR95 where it first appears. Also, the definition in L415 can be improved by stating that it uses a threshold such that 95% of ID samples are correctly classified.
- The caption of Figure 5 needs improvement; it is currently unclear which histogram corresponds to which method. Similar clarification should be added for Figure 2 for the attractor reconstructions, by specifying whether those are latent attractors or decoded outputs.

**Questions:**

- For a pretrained model, is it possible to generate attractors from random noise and gain intuition on whether the model is operating in the generalization or memorization regime?
- Do the reconstructions of attractors for the pretrained models carry interpretable or semantically meaningful information?

---

> ### Author Response · Authors · 2025-11-23
> **Rebuttal (1 of 3)**
>
> We thank the Reviewer for their useful feedback and suggestions and for appreciating the paper. We address their questions and concerns below and remain available for additional clarification in the discussion period.
>
>
> ### **Latent vector fields beyond autoencoders**.
>
> We agree that assessing whether attractors exist in forecasting or next-token predictors, as well as encoder-only architectures, would strengthen the impact. Our theory (Theorem 1 and Proposition 3.1) does not strictly require an autoencoder architecture, but only a **self-map that is locally contractive**.
> To preliminarily study whether the framework extends beyond autoencoders, we evaluated two settings, which we included fully detailed in Appendix B.8 of our revised manuscript. In summary, we considered:
>
> 1. **Self-supervised encoders (e.g., DINOv2, SigLIP2)** using decoders on top of the frozenencoder model. Iterative application of the induced latent map shows **bounded norms and vanishing reconstruction/latent residuals**, indicating well-behaved contraction dynamics.
>
> 2. **Next token predictors-LLMs (e.g., Qwen3-0.6B, SmolLM2)** by iterating hidden representations through the residual stream. Both models show **contractive trajectories without divergence**, again yielding stable latent vector fields.
> Taken together, these results  preliminary demonstrate that latent vector fields emerge across architectures beyond autoencoders. Although further analysis is needed to characterize attractors and architectural influences on the dynamics we believe that this framework is promising to be applied to general models in future work
>
> We also gently note that autoencoders remain widely used: for example denoising autoencoders are building blocks in diffusion models (e.g. [d]), in masked autoencoding for vision (e.g. [a,f,d]) and biological applications (e.g. [b]), and in sparse autoencoders (e.g.[c,e]) used in mechanistic interpretability of large foundation models. We believe that using our framework in this setting could help understanding better properties of these models.
>
>
> _[a] Xiang, et al. "Denoising diffusion autoencoders are unified self-supervised learners." CVPR 2023._
>
> _[b]Kraus, et al. "Masked autoencoders for microscopy are scalable learners of cellular biology." CVPR 2024._
>
> _[c] Cunningham et al. "Sparse autoencoders find highly interpretable features in language models." ICLR 2024_
>
> _[d] Chen, et al. "Masked autoencoders are effective tokenizers for diffusion models." ICML 2025._
>
> _[e] Gao, et al. "Scaling and evaluating sparse autoencoders." ICLR 2025_
>
> _[f] Wang, et al. "Videomae v2: Scaling video masked autoencoders with dual masking." CVPR 2023._
>
> ---
>
> ### **What do attractors correspond to in input space?**
>
>  Attractors coincide with training examples when the model operates in a memorization regime (see Fig. 2a, first row). In the generalization regime, attractors correspond to nonlinear combinations of training samples (Fig. 2a, last rows) and could also correspond to sets as opposed to discrete points. For instance, in Sec. 4.1 the Stable Diffusion autoencoder operates in the *generalization* regime, and attractors do not map directly to training samples. We include additional qualitative examples in the Appendix B.9.  For models in the *memorization* regime,  training examples can be recovered in principle even when starting from noise (provided enough samples are drawn).
>
> ---
>
> ### **Local contractivity assumption**
>
> We reported a convergence analysis on the latent trajectories computed in ViT-MAE for the experiment in section 4.2, in Appendix B.4.1 (Figure 12).  Specifically we perform the convergence analysis it on trajectories from 2000 samples drawn from Imagenet, showing that residual error in the latent spaces goes to zero demonstrating contraction behavior. We also report the norms of the embeddings at each iteration showing that these are bounded, excluding any divergent behavior.
>
> This behavior is consistent with what is measured in Appendix B.8, with the difference that residual error there stabilizes without going to zero, and with the experiments on smaller autoencoders trained on MNIST, F-MNIST, CIFAR10. (see for example Figure 1).
>
> In general in our experiments we **never** observed divergent behavior of the latent vector fields.
>
> ---

---

> > ### Author Response · Authors · 2025-11-23
> > **Rebuttal (2 of 3)**
> >
> > ### **Comparison with OOD baselines**
> >
> > We thank the reviewer for the suggestion. We compared the trajectory-based scoring with the KNN, Mahalanobis distance in latent space, and reconstruction error baselines below. Results confirms that trajectory statistics are informative for distribution shift:
> >
> > | Method              | SUN397 FPR ↓ | SUN397 AUROC ↑ | Places365 FPR ↓ | Places365 AUROC ↑ | Texture FPR ↓ | Texture AUROC ↑ | iNaturalist FPR ↓ | iNaturalist AUROC ↑ |
> > |--------------------|--------------|----------------|------------------|-------------------|---------------|------------------|--------------------|---------------------|
> > | *KNN*              | 100.0        | 42.6           | 100.0            | 32.4              | 34.5          | 89.4             | 86.4              | 68.6               |
> > | *Mahalanobis*      | 82.0         | 58.0           | 88.0             | 45.0              | 43.0          | 90.0             | 31.0               | 87.0               |
> > | *Reconstruction*   | 91.8         | 49.5           | 91.6             | 50.9              | 98.4          | 49.1             | 99.2               | 23.6               |
> > | *d(Attractors)*     | **29.6**     | **91.2**       | **29.9**         | **91.0**          | **25.9**      | **92.6**         | **29.9**           | **91.3**            |
> >
> >
> > In general we believe that latent trajectory and attractors statistics could boost the performance of feature based OOD detection methods as latent trajectories contain the features themselves (which correspond to the trajectory at time t=0) therefore they should be strictly more informative. As an example, classification based methods could make use of the entire trajectory towards attractors to classify samples rather than the features alone. We leave the exploration of this for future work.
> >
> >
> > ---
> >
> > ### **KNN analysis details**
> >
> > For the KNN baseline, for each test point we compute the K=2000 nearest neighbors from ImageNet validation samples (ID set), and the reported score is the mean distance over this neighborhood. This follows [g] without the contrastive fine-tuning stage. We clarified the baseline and added the reference in the paper.
> >
> > **Sensitivity to K is shown below**; results get better when considering more local neighborhoods, but still distance of **latent trajectories to attractor outperform these settings**.
> >
> >
> > |  Method / K        | SUN397 FPR ↓ | SUN397 AUROC ↑ | Places365 FPR ↓ | Places365 AUROC ↑ | Texture FPR ↓ | Texture AUROC ↑ | iNaturalist FPR ↓ | iNaturalist AUROC ↑ |
> > |--------------------|--------------|----------------|------------------|-------------------|---------------|------------------|--------------------|---------------------|
> > | 1                  | 100.0 | 0.1 | 100.0 | 0.1 | 100.0 | 0.1 | 100.0 | 0.1 |
> > | 5                  | 92.5  | 48.7 | 94.1  | 42.2 | 33.2  | 89.8 | 30.7  | 90.0 |
> > | 10                 | 94.6  | 45.5 | 96.0  | 37.9 | 37.6  | 89.6 | 31.5  | 88.0 |
> > | 20                 | 95.9  | 43.4 | 97.4  | 34.6 | 42.6  | 89.4 | 33.0  | 86.0 |
> > | 30                 | 96.5  | 42.5 | 97.8  | 33.1 | 45.0  | 88.7 | 34.8  | 84.8 |
> > | 50                 | 97.4  | 41.6 | 98.2  | 31.3 | 48.0  | 88.0 | 37.9  | 83.5 |
> > | 100                | 97.5  | 40.5 | 98.7  | 29.1 | 49.9  | 87.1 | 42.4  | 81.7 |
> > | 200                | 97.6  | 39.7 | 98.9  | 27.4 | 54.2  | 86.2 | 46.5  | 79.8 |
> > | 300                | 97.6  | 39.4 | 98.9  | 26.7 | 55.9  | 85.7 | 48.6  | 78.6 |
> > | 500                | 97.6  | 39.4 | 98.8  | 26.2 | 59.3  | 85.0 | 51.3  | 77.1 |
> > | 1000               | 96.9  | 40.0 | 98.7  | 26.2 | 61.8  | 84.1 | 56.0  | 74.8 |
> > | 1500               | 96.2  | 40.7 | 98.4  | 26.7 | 63.2  | 83.6 | 58.7  | 73.4 |
> > | 2000               | 100.0 | 42.6 | 100.0 | 32.4 | 34.5  | 89.4 | 86.4  | 68.6 |
> > | *d(Attractors)*    | **29.6** | **91.2** | **29.9** | **91.0** | **25.9** | **92.6** | **29.9** | **91.3** |
> >
> >
> >
> > We will include this ablation in the Appendix. For our trajectory based score, we consider the average to the 2000 attractors computed from the 2000 sampled training points: restricting to K nearest attractors could be applicable to our setting as well and we leave the exploration for future work.
> >
> >
> > _[g] Sun, Yiyou, et al. "Out-of-distribution detection with deep nearest neighbors." International conference on machine learning. PMLR, 2022._
> >
> > ---

---

> ### Author Response · Authors · 2025-11-23
> **Rebuttal (3 of 3)**
>
> ### **Performance under gaussian noise**
>
> We recomputed and present results below: we observe that performance disrupts indicating that the attractors computed from noise are less informative to . Performance however is still better than using the reconstruction error as a score indicating the presence of weak signals.  This may indicate either that there is very small overlap between attractors from noise and data in the ViT-MAE model but also sensitivity to the sampling.
>
>
>
> | Method                  | SUN397 FPR ↓ | SUN397 AUROC ↑ | Places365 FPR ↓ | Places365 AUROC ↑ | Texture FPR ↓ | Texture AUROC ↑ | iNaturalist FPR ↓ | iNaturalist AUROC ↑ |
> | ----------------------- | ------------ | -------------- | --------------- | ----------------- | ------------- | --------------- | ----------------- | ------------------- |
> | *d(Attractors ; Noise)* | 86.4         | 48.4           | 85.9            | 49.7              | 83.1          | 54.8            | 85.7              | 49.6                |
> | *d(Attractors; Data)*  | **29.6**     | **91.2**       | **30.0**        | **91.0**          | **25.9**      | **92.6**        | **29.9**          | **91.3**            |
>
>
>
> ### **Definition of trajectory-based scoring**
>
> The trajectory-based score is defined as the mean distance across attractors (analogous to Sec. 4.2). We will clarify this in the manuscript.
>
> ---
>
>
>
> ### **Pretrained models: intuition about generalization vs memorization**
>
> For pretrained models, attractors can be computed from noise to probe whether the model is in a memorization or generalization regime. Whether it is possible to estimate properties about the training distribution without accessing a single data point, this depends on how much the probability distribution of the attractors computed from noise overlaps with the one of the training data. In Figure 3 (d) we measured the similarity between the two distributions, showing that for a model in generalization regime, noise attractors show high similarity to training ones (despite the latent trajectories being different).
> As a future direction we believe that the geometry and topology of attractors (i.e. being discrete points or continuous manifold) could inform on a model operating in the memorization of generalization regime,
> In practice, for large models, we believe it might be harder to draw conclusions without having access to any training data point (or information on the training distribution)  due to the number of samples required in order to probe the model.
>
>
> ---
>
> ### **FPR95 definition and caption clarity**
>
> We clarified the definition of FPR95 where first introduced and updated captions for Figures 2 and 5 to specify which histograms correspond to which method, and whether reconstructions refer to latent attractors or decoded outputs.
>
>
> ---
>
> ### **Semantic information of attractors**
>
> We didn’t observe semantic information contained in decoded attractors in Stable diffusion or ViT-MAE. These resemble more geometrical patterns and textures, similarly as for the case on MNIST in the last row of Figure 2 (c). We included visualizations in Appendix B.9

---

> > ### Comment · Reviewer_XuAT · 2025-11-24
> >
> > I thank the authors for their detailed responses and additional analyses. I don't think that the results on LLMs and self-supervised models clearly tell us anything about memorization vs. generalization (which would be a very impactful result if shown). But I agree this is a reasonable first step toward studying their latent dynamics. I also understand that doing such an analysis would require training control models that we know lie in memorization or generalization regimes, which is computationally expensive.
> >
> > Overall, it still seems that analyzing the generalization behavior of pretrained models with this method may require access to the original pretraining data—unless future work can provide stronger evidence like the analysis I mentioned. That said, the idea itself is interesting and promising, and I see it as a solid first step in that direction. I therefore recommend acceptance.

---

> ### Author Response · Authors · 2025-11-26
>
> We thank the reviewer for following up and for raising the score. We fully agree that assessing generalization or memorization, both in non–autoencoder models and/or without access to the training set, is an exciting and impactful direction.
>
> In this regard we believe that the geometry of attractors, in particular, whether the dynamics converge to discrete points or to higher-dimensional attractor manifolds and their distribution in the latent space, may provide useful signals about these regimes and help relax the need for training-data access. Developing these ideas is a natural next step, and we appreciate the reviewer's suggestions in this regard.
>
> We remain available for any further questions for the reminder of the discussion period.

---

### Official Review · Reviewer_LZxN · 2025-10-28

**Soundness:** 3
**Presentation:** 4
**Contribution:** 3
**Rating:** 6
**Confidence:** 3

**Summary:**

This work introduced a method to interpret autoencoder neural networks as dynamical systems defined by a latent vector field on their manifold. This vector field on their manifold is derived by iteratively applying their encoding-decoding map.

The paper claims that inductive biases introduced by standard training can be seen as emerging attractor points in their latent vector fields and propose to leverage this vector fields as representations of the neural network for downstream tasks such as (i) the analysis of the neural network with respect to generalization and memorization, (ii) the extraction of knowledge encoded in the weights of the neural network, and (iii) as tool to identify out-of-distribution samples.

The paper presents three experiments. The first experiment investigates the relationship between generalization and memorization and the role of regularization on 30 convolutional AE trained on small-scale datasets such as CIFAR10, MNIST, and FashionMNIST. The second experiment aims to investigate vision foundation models and probe the recovery of information about the data encoded in the models' weights. This is done on Stable diffusion AE and vision transformer masked AEs. The third experiment aims to demonstrate the method's expressiveness to detect distribution shifts of input data from the latent trajectories of the vector field.

I like the idea and think that this paper does well in motivating the proposed methods, providing a theoretical foundation for it, and demonstrating the method's utility through the set of downstream tasks. Unfortunately, this approach is limited to encoder-decoder models, which is mentioned in the limitation section of the papers.

There is one open question that I would appreciate getting answered by the authors. There exists another method that learns a lower-dimensional manifold of neural network models using an autoencoder architecture. The embeddings on this manifold are then used for several downstream tasks, revealing the encoded information of the neural network's weights. This paper and the method I have mentioned sound similar, and I would like to make sure that they are different, as I understand it. I do provide mode details in the questions section.

**Strengths:**

- **(S1)**: I appreciate the paper's motivation and theoretical foundation. It provides an interesting view of (AE) neural networks and provides a novel tool for analysis.

- **(S2)**: I think that the experimental section is honestly aiming to demonstrate the method's utility with respect to different downstream tasks and different datasets. I also appreciate the details and additional results listed in the appendix.

**Weaknesses:**

- **(W1)**: The proposed method is limited to reconstruction-based autoencoder neural networks. The authors are aware of this as they do mention this in the imitation section.

**Questions:**

- **(Q1)**: As mentioned above, I would like to make sure I understand the presented approach properly and do not confuse it with another method. In [1], a lower-dimensional manifold of neural network weights is learned by an encoder-decoder setup using a reconstruction loss and a self-supervised loss. This encoder-decoder bottleneck is interpreted as the representation of a neural network, which itself can be used to reveal encoded information of the neural network. I think that this submission is different from the work mentioned, but I am not sure since the methods are similar in their terminology and ideas.

[1] Self-Supervised Representation Learning on Neural Network Weights for Model Characteristic Prediction
Schuerholt et al, NeurIPS, 2021

**Details Of Ethics Concerns:**

-

---

> ### Author Response · Authors · 2025-11-23
> **Rebuttal**
>
> We thank the Reviewer for their useful feedback and suggestions and for appreciating the paper. We address their questions and concerns below and remain available for additional clarification in the discussion period.
>
>
> ### **Beyond autoencoders**
>
>  We thank the reviewer for raising this point. While we focused on autoencoders, we would like to gently note that our theory (Theorem 1 and Proposition 3.1) does not strictly require an autoencoder architecture, but only a **self-map that is locally contractive**. We will clarify this in the paper.
> We agree that assessing whether attractors  and latent dynamics can be extended beyond autoencoders (for example for next-token predictors, as well as encoder-only architectures), would strengthen the impact. To preliminarily study whether the framework extends beyond autoencoders, we included novel experiments  evaluating two settings, which we included fully detailed in Appendix B.8 of our revised manuscript. In summary, we considered:
>
> 1. **Self-supervised encoders (e.g., DINOv2, SigLIP2)** using decoders on top of the frozenencoder model. Iterative application of the induced latent map shows **bounded norms and vanishing reconstruction/latent residuals**, indicating well-behaved contraction dynamics.
>
> 2. **Next-token LLMs (e.g., Qwen3-0.6B, SmolLM2)** by iterating hidden representations through the residual stream. Both models show **contractive trajectories without divergence**, again yielding stable latent vector fields.
> Taken together, these results  preliminary demonstrate that latent vector fields emerge across architectures beyond autoencoders. Although further analysis is needed to characterize attractors and architectural influences on the dynamics we believe that this framework is promising to be applied to general models in future work
> We also gently note that autoencoders remain widely used: for example denoising autoencoders are building blocks in diffusion models (e.g. [d]), in masked autoencoding for vision (e.g. [a,f,d]) and biological applications (e.g. [b]), and in sparse autoencoders (e.g.[c,e]) used in mechanistic interpretability of large foundation models. We believe that using our framework in this setting could help understanding better properties of these models.
>
>
>
> _[a] Xiang, et al. "Denoising diffusion autoencoders are unified self-supervised learners." CVPR. 2023._
>
> _[b]Kraus, et al. "Masked autoencoders for microscopy are scalable learners of cellular biology." CVPR. 2024._
>
> _[c] Cunningham et al. "Sparse autoencoders find highly interpretable features in language models." ICLR 2024_
>
> _[d] Chen, et al. "Masked autoencoders are effective tokenizers for diffusion models." ICML. 2025._
>
> _[e] Gao, et al. "Scaling and evaluating sparse autoencoders." ICLR 2025_
>
> _[f] Wang, et al. "Videomae v2: Scaling video masked autoencoders with dual masking." CVPR. 2023._
>
> ---
>
> ### **Relation to [1]**
>
> Thanks for pointing out this connection. We agree with the Reviewer's take and confirm that our work is quite distinct from [1].
> In [1], the autoencoder is explicitly trained to encode neural network weights: the encoder–decoder takes model parameters as inputs and learns a low-dimensional embedding of weights, effectively acting as a hypernetwork. As the reviewer noted, this embedding is then used to study information encoded in network parameters.
> In our case, the latent vector field is not trained to encode other models, nor does it require any additional objective or dataset. It is simply obtained by iterating the autoencoder map in the latent space of a single data-trained autoencoder, revealing fixed points and attractor structure induced by the representation itself. The focus is on latent dynamics of representations learned from data, not embeddings of model weights.
> That said, applying our framework to hypernetwork autoencoders trained on weights (such as [1]) could be a compelling direction to study the geometry of neural-network parameter spaces, and we have now included this reference in the conclusion as a promising avenue for future work.
>
> _[1] Self-Supervised Representation Learning on Neural Network Weights for Model Characteristic Prediction Schuerholt et al, NeurIPS, 2021_

---

> > ### Author Response · Authors · 2025-11-28
> >
> > We thank again the reviewer for their comments and suggestions. We hope to have addressed all their concerns and questions in our rebuttal, also in light of the positive replies from the rest of the reviewers and reviewers b56C and XuAT increasing their scores.
> >
> > We remain available for the reminder of the discussion period for any additional clarifications.

---

### Official Review · Reviewer_b56C · 2025-10-29

**Soundness:** 3
**Presentation:** 3
**Contribution:** 2
**Rating:** 6
**Confidence:** 4

**Summary:**

The paper presents an alternative representation of autoencoder (AE) models as dynamical systems acting on the latent manifold. The authors introduce the theoretical framework required for this interpretation and establish results linking the dynamical system induced by the iterative application of an AE to the gradients of the underlying data distribution. They also theoretically characterize the attractors of the latent vector field and link them to memorization and generalization.

This framework is then used to analyze how regularization affects the phenomena of memorization and generalization in AEs, showing that as memorization decreases and generalization increases, the attractors of the AEs evolve from latent points corresponding to training examples toward more general attractors. The paper further shows that a transition from memorization to generalization occurs during training, with an increasing number of attractors being learned and the similarity of attractors found using different data converging.

Finally, the authors extract AEs from common pre-trained models and show that noisy inputs can be used to find the attractors of the induced dynamical system, and that these attractors can serve as a dictionary helping reconstruct data points from diverse distributions (as compared with a random orthogonal basis). The trajectories of examples can also be used for OOD detection.

**Strengths:**

- This new perspective on AEs is simple and intuitive. It is somewhat surprising that this type of analysis has not been done sooner.
- The links to regularization, memorization, and generalization are interesting, and the proposed framework could be a useful analysis tool.
- The theoretical framework is well presented and clear, and the theoretical results appear correct.
- The paper is well written; the dynamical-systems terminology is clear and intuitive.
- The experiments using AEs extracted from pre-trained models are particularly strong, without these, the toy settings described earlier would not have been sufficient.

**Weaknesses:**

- The scope of the paper is somewhat limited since the theory only holds for AEs.
- While the proposed framework is well justified and interesting in its own right, its impact is difficult to gauge. There is no immediate practical impact for practitioners, nor any strikingly new finding that this framework helps uncover. However, the work has clear potential as a future analysis tool.

See the Questions section for more precise comments.

**Questions:**

- Theorem 1 strength and scope: The result relies on uniform contractivity and latent concentration on fixed points, which are strong assumptions rarely satisfied by general AEs. The empirical section provides motivation for approximate contractivity, but the theorem should be reframed as a local or heuristic alignment with the score, not strict proportionality.
- Banach fixed-point misstatement: The text says well-posedness “holds iff f is Lipschitz-continuous.” Banach’s theorem requires a contraction (Lipschitz constant < 1), not mere Lipschitz continuity. Please correct.
- Definition 3 typo: There appears to be an extra f after the Jacobian when defining the Lipschitz constant in Definition 3.
- Detail: The numbering of the Theorems / Propositions is inconsistent across the main paper and appendix, which is confusing. Either match the numbering exactly or link the appendix theorem/proposition in the main paper.
- Section 4.1: The claim that “OMP = PCA” when using a random orthogonal basis is incorrect. PCA involves the data-covariance eigenbasis; OMP on a random orthobasis is simply sparse projection in that basis, not PCA. Please fix the description and, ideally, compare against true PCA (top-k principal components learned from data) as a stronger baseline.
- From the definition of the trajectory score in Sec. 3.2.2, it is not directly clear whether the distance is to all training attractors at once, and which exact point-cluster distance is used (this likely affects results since different point-cluster distances capture different notions of similarity).
- OOD baselines are too weak. The proposed trajectory-distance score is only compared to K-NN (with K = 2000). Modern OOD detection for vision backbones includes MSP/energy scores, Mahalanobis, ViM, ODIN, KLM, etc. Adding these would materially strengthen the claim that trajectories convey additional signal beyond embeddings. Also, which neighbors are considered? Since it is trajectories that are analyzed, and each embedding along the trajectory may have different K-NNs, the reference points are moving if you recompute the K-NN for each point in the trajectory.
- The distinction between “aggressive regularization” (1) and “over-parameterization” (2) as two forms of memorization is very interesting and would warrant further analysis. Perhaps this framework could allow the characterization of different forms of memorization in NNs. Currently, these are disjoint and hard to compare since (1) is presented in the main text as a function of k (latent-space dimension) and (2) is presented in the appendix as a function of dataset size. Unifying these observations would be valuable.

---

> ### Author Response · Authors · 2025-11-23
> **Rebuttal (1 of 2)**
>
> We thank the Reviewer for their useful feedback and suggestions and for appreciating the paper. We address their questions and concerns below and remain available for additional clarification in the discussion period.
>
> ### **Scope of the paper**
> We gently note that Theorem 1 and proposition 3.1 generally hold for **any self map which is locally contractive**, not only autoencoders, we are going to clarify this in the paper.
> In the paper we decided to focus on the autoencoder case given its natural fit to the setting, but we agree that providing perspective for applicability of our framework beyond autoencoders is important. To this end **we provide preliminary evidence** that latent vector fields and contractive iterative dynamics **are not unique to autoencoders**. We evaluate two additional settings:
> 1. **Self-supervised encoders (e.g., DINOv2, SigLIP2)** using pretrained decoders. Iterative application of the induced latent map shows **bounded norms and vanishing reconstruction/latent residuals**, indicating well-behaved contraction dynamics.
> 2. **Next-token LLMs (e.g., Qwen3-0.6B, SmolLM2)** by iterating hidden representations through the residual stream. Both models show **contractive trajectories without divergence**, again yielding stable latent vector fields.
>
> Taken together, these results preliminary show that latent vector fields emerge across architectures beyond autoencoders, making latent vector fields a useful tool to study general neural models, not particularly restricted to a single architectural or modelling choice.
>
> We also gently note that autoencoders remain widely used: for example denoising autoencoders are building blocks in diffusion models (e.g. [d]), in masked autoencoding for vision (e.g. [a,f,d]) and biological applications (e.g. [b]), and in sparse autoencoders (e.g.[c,e]) used in mechanistic interpretability of large foundation models. We believe that our framework could help understanding better the properties of such models.
>
> _[a] Xiang, et al. "Denoising diffusion autoencoders are unified self-supervised learners." CVPR 2023._
>
> _[b]Kraus, et al. "Masked autoencoders for microscopy are scalable learners of cellular biology." CVPR 2024._
>
> _[c] Cunningham et al. "Sparse autoencoders find highly interpretable features in language models." ICLR 2024_
>
> _[d] Chen, et al. "Masked autoencoders are effective tokenizers for diffusion models." ICML 2025._
>
> _[e] Gao, et al. "Scaling and evaluating sparse autoencoders." ICLR 2025_
>
> _[f] Wang, et al. "Videomae v2: Scaling video masked autoencoders with dual masking." CVPR 2023._
>
> ---
>
> ###  **Clarification on KNN.**
>
> For the KNN baseline, for each test point we compute the K-NN (with K=2000) neighbors from samples of the validation set of ImageNet (i.e., the in distribution (ID)  set). Then for the score we compute the mean over this distribution. The baseline resembles the approach of [g], without the contrastive fine tuning stage. We added the reference in the main paper.
>
> ### **Additional OOD baselines.**
>
>  We gently remark that suggested baselines (Energy, MSP,..) use information on ID labels (either through logits, or weights of the classifier or class centroids), while both our method and the KNN baseline are completely **unsupervised**, and do not require access to labels, which we believe is more fitting to the autoencoder setting. To this end, we added as comparison the reconstruction error of the autoencoder and mahalonabis distance, presented in the Table below.  We also remark that the KNN baseline was already pointed out in [g] to compete with or outperform suggested baselines for comparison, in the supervised setting.
> Results **confirm that using the latent trajectories attractors statistics is informative for capturing distribution shifts**.
>
>
> | Method              | SUN397 FPR ↓ | SUN397 AUROC ↑ | Places365 FPR ↓ | Places365 AUROC ↑ | Texture FPR ↓ | Texture AUROC ↑ | iNaturalist FPR ↓ | iNaturalist AUROC ↑ |
> |--------------------|--------------|----------------|------------------|-------------------|---------------|------------------|--------------------|---------------------|
> | *KNN*              | 100.0        | 42.6           | 100.0            | 32.4              | 34.5          | 89.4             | 86.4              | 68.6               |
> | *Mahalanobis*      | 82.0         | 58.0           | 88.0             | 45.0              | 43.0          | 90.0             | 31.0               | 87.0               |
> | *Reconstruction*   | 91.8         | 49.5           | 91.6             | 50.9              | 98.4          | 49.1             | 99.2               | 23.6               |
> | *d(Attractors)*     | **29.6**     | **91.2**       | **29.9**         | **91.0**          | **25.9**      | **92.6**         | **29.9**           | **91.3**            |
>
>
> _[g] Sun, Yiyou, et al. "Out-of-distribution detection with deep nearest neighbors." International conference on machine learning. PMLR, 2022._

---

> ### Author Response · Authors · 2025-11-23
> **Rebuttal (2 of 2)**
>
> ### **Additional OOD baselines (contd)**
>
> In general we gently remark that the goal of this experiment is to **show that latent trajectories contain signals about distribution shift**, as opposed to proposed a novel OOD detection algorithm, which could be pursued as future work.
>
> We believe that OOD detection methods could benefit from using the information in the latent trajectory as this contains the features themselves (which correspond to the trajectory at time t=0) therefore it should be **strictly more informative**. As an example, classification based methods could make use of the entire trajectory towards attractors to classify samples rather than the features alone. We leave the exploration of this for future work.
>
>
> ---
>
> ### **Theorem 1 strength and scope**.
>
> We agree that assumptions in Theorem 1 may not be fully met in practice, however we also gently remark in the paper that many inductive biases (initialization, impolite and explicit regularization) enforce models to be locally contractive. **We added a precise connection between weight decay and the norm of the Jacobian in Appendix C** as an example and we showed multiple examples in Table 2 in the appendix.  This is also supported by our experiments where we **never** encountered divergent trajectories in the latent vector fields. The purpose of Theorem 1 we believe is to show what consequences a contraction has on the latent vector field, even if its realization might be approximated in practice.  We agree that the statement in the main paper needs to be clear and **we rephrased specifying that the alignment with score is local**.
>
> ---
>
> ### **OMP PCA experiment.**
>
>  We clarified the description in the paper. We gently remark that both the orthogonal basis and the attractors are computed from noise and **do not use any knowledge about the data**, therefore computing a PCA basis from data to perform the same experiment would not be a fair baseline.  However to understand how it would perform, we ran an experiment computing principal components from 8000 latent samples of LAION dataset and reported results below: the main observation is that despite performing better than the orthogonal basis, the attractors approach still outperforms the PCA baseline.
>
> | Dataset | Method | 32 | 64 | 128 | 256 | 512 | 1024 | 2048 |
> |---|---|---|---|---|---|---|---|---|
> | LAION-2B | **Ours** | **0.29 ± 0.12** | **0.25 ± 0.10** | **0.22 ± 0.09** | **0.17 ± 0.07** | **0.13 ± 0.06** | **0.09 ± 0.05** | **0.06 ± 0.04** |
> |  | PCA | 0.43 ± 0.23 | 0.38 ± 0.19 | 0.33 ± 0.16 | 0.26 ± 0.11 | 0.18 ± 0.09 | 0.13 ± 0.07 | 0.08 ± 0.05 |
> | CIFAR-100 | **Ours** | **0.01 ± 0.01** | **0.03 ± 0.01** | **0.05 ± 0.02** | **0.08 ± 0.04** | **0.11 ± 0.05** | **0.13 ± 0.06** | **0.16 ± 0.08** |
> |  | PCA | 0.05 ± 0.03 | 0.09 ± 0.05 | 0.13 ± 0.05 | 0.16 ± 0.07 | 0.19 ± 0.09 | 0.21 ± 0.12 | 0.24 ± 0.14 |
> | ImageNet-1K | **Ours** | **0.05 ± 0.04** | **0.08 ± 0.04** | **0.10 ± 0.04** | **0.12 ± 0.05** | **0.14 ± 0.06** | **0.16 ± 0.06** | **0.18 ± 0.07** |
> |  | PCA | 0.07 ± 0.03 | 0.10 ± 0.05 | 0.14 ± 0.05 | 0.17 ± 0.06 | 0.20 ± 0.08 | 0.23 ± 0.09 | 0.26 ± 0.11 |
> | PatchCamelyon | **Ours** | **0.09 ± 0.05** | **0.13 ± 0.06** | **0.17 ± 0.06** | **0.22 ± 0.05** | **0.28 ± 0.05** | **0.31 ± 0.06** | **0.36 ± 0.08** |
> |  | PCA | 0.15 ± 0.05 | 0.21 ± 0.07 | 0.23 ± 0.05 | 0.27 ± 0.06 | 0.31 ± 0.09 | 0.35 ± 0.12 | 0.38 ± 0.16 |
> | EuroSAT | **Ours** | **0.00 ± 0.00** | **0.01 ± 0.00** | **0.02 ± 0.01** | **0.04 ± 0.02** | **0.05 ± 0.03** | **0.06 ± 0.03** | **0.07 ± 0.04** |
> |  | PCA | 0.01 ± 0.01 | 0.02 ± 0.01 | 0.04 ± 0.01 | 0.05 ± 0.02 | 0.06 ± 0.03 | 0.07 ± 0.04 | 0.08 ± 0.05 |
> | Places365 | **Ours** | **0.07 ± 0.03** | **0.10 ± 0.03** | **0.12 ± 0.04** | **0.14 ± 0.04** | **0.16 ± 0.04** | **0.18 ± 0.05** | **0.20 ± 0.05** |
> |  | PCA | 0.10 ± 0.03 | 0.14 ± 0.04 | 0.17 ± 0.04 | 0.20 ± 0.05 | 0.23 ± 0.06 | 0.25 ± 0.06 | 0.28 ± 0.07 |
>
>
> ---
>
> ### **Aggressive regularization and overparametrization**.
>
> We thank the reviewer for the insightful comment. We agree that the point is very interesting, our view is the following: strong regularization induces underfitting, while extreme overparametrization may induce overfitting. Generalization lies between these regimes. When both solutions memorize similar training points, they may differ in regions between attractors: aggressive regularization yields smoother basins, while overparametrized solutions may create sharper ones. We will clarify this perspective in the paper.
>
> ---
>
> **Clarification on  of trajectory score (sec 3.2.2).**: The distance is defined as mean distance over training attractors (akin to the experiment in section 4.2) .  We do not compute attractors from all training points but we sample from the training set.
>
> ---
>
> **Typos and polishing**: we thank the reviewer for spotting typos and imprecisions: we fixed them in the paper, namely: we addressed the Banach’s theorem statement and typo in definition 3 and PCA description.

---

> > ### Comment · Reviewer_b56C · 2025-11-26
> >
> > I thank the authors for their detailed and thoughtful rebuttal, as well as for the substantial updates made to the manuscript. Most of my concerns have been properly addressed. In particular, I appreciate the clarifications to the theoretical statements, the corrections of technical inaccuracies, the strengthened PCA and OOD baselines, and the added analyses regarding contractivity and trajectory behavior. These additions considerably improve both the rigor and clarity of the work.
> >
> > A few points raised in the response appear to be planned but not yet fully incorporated (e.g., clarifications related to applicability beyond autoencoders, additional discussion on the relationship between different forms of memorization). I expect the authors to integrate any missing experiments or analysis and ensure that all explanations provided in the rebuttal are clearly reflected in the final version of the manuscript.
> >
> > Overall, the revision meaningfully strengthens the contribution and addresses the main concerns I raised. I am happy to increase my score to 8.

---

> ### Author Response · Authors · 2025-11-28
>
> We thank the reviewer for following up and for increasing the score.
>
> We will include all the remaining experiments and discussion added during the rebuttal in the paper.
>
> We thank again the reviewer for all their suggestions that contributed to strengthen the paper.
>
> We remain available in case of further questions for the reminder of the discussion period.

---

### Official Review · Reviewer_KKEL · 2025-10-31

**Soundness:** 3
**Presentation:** 3
**Contribution:** 3
**Rating:** 8
**Confidence:** 3

**Summary:**

This paper shows that iterating the autoencoder‑induced map $f(z)=E \circ D(z)$ implicitly defines a vector field in latent space. It then exploits the field's dynamics and attractor structure to diagnose memorization versus generalization and to detect out‑of‑distribution (OOD) inputs.

**Strengths:**

- It is a novel and distinctive observation that iteratively applying $f$ induces the residual vector field $V(z) = f(z)-z$, whose fixed points serve as attractors toward which nearby trajectories converge.
- The claim that this vector field is proportional to the score of the latent prior $q(z)$ is highly intriguing; it effectively generalizes the small‑noise limit result for denoising autoencoders to the latent space.
- Proposition 2 is particularly insightful: when training biases the model toward memorization, the prototype term approaches zero while the coverage term narrows, yielding a clear, interpretable criterion for judging memorization versus generalization from the proposed error decomposition.
- The paper also establishes a lower bound on the number of iterations required to converge in simple linear settings, grounding the dynamics with an interpretable complexity estimate.

**Weaknesses:**

- The explanation for why contraction emerges *naturally*  via initialization bias, explicit regularization, and implicit regularization would benefit from a stronger theoretical foundation or, at least, a more formal set of sufficient conditions.
- Several assumptions, e.g., smoothness of the induced latent distribution and related regularity, are stated, but the extent to which they hold for large‑scale models in practice remains unclear.

**Questions:**

- **Numerical validation of Theorem 2.** Can you empirically validate Theorem 2? Such evidence would help assess the plausibility of the assumptions underlying its derivation and test the theorem's robustness in realistic settings.
- **Iteration complexity under weaker assumptions.** Is it possible to analyze (or bound) the number of iterations required to reach a fixed point under assumptions weaker than those currently stated?

---

> ### Author Response · Authors · 2025-11-23
> **Rebuttal (1 of 2)**
>
> We thank the Reviewer for their useful feedback and suggestions and for appreciating the paper. We address their questions and concerns below and remain available for additional clarification in the discussion period.
>
> ### **Relation between contraction and regularization**
>
> Thanks for raising this point.  We expand the discussion below and include additional examples and derivations in **Appendix C**.
>
> We start by gently remarking that Theorem 1 assumes that the network is locally contractive around the training examples. For autoencoders this holds whenever the learned mapping corresponds to the minimum solution of:
>
> $$
> \mathbb{E}_{x \sim p{X}} \left[ | D(E(x)) - x |^2 + \lambda |J_f(E(x))|_F^2 \right], \lambda > 0.
> $$
>
> Our argument in the paper is that initialization bias, explicit regularization, and implicit regularization frequently *promote* contractive behavior in practice, even if they do not directly optimize for a penalty on the Jacobian norm.  Below we clarify further the connection and give concrete settings where contractivity can be derived.
>
> * *Weight decay.* For some architectures and regularizers, the link with contractivity is direct. For example in a linear model $(F(x) = Wx)$ , the Jacobian is $J_f(x) = W .$ Weight decay reduces $(|W|)$, making it the primary mechanism that enforces contraction. For feedforward nonlinear network with 1-Lipschitz activation (e.g. ReLU, GELU, SILU, tanh)
> $$
> F_\theta(x) = W_L,\phi_{L-1} \left(\cdots \phi_1(W_1 x)\right),
> $$
> the Jacobian satisfies (by the chain rule and submultiplicativity):
> $$
> J_\theta(x) = W_L D_{L-1}(x)\cdots D_1(x) W_1,
> $$
>
>
> $$
> \|J_{\theta}(x)\|_2 \le |W_{L}|_2,|D_{L-1}(x)|_2 \cdots |W_1|_2 \le \prod_{l=1}^L |W_l|_2 ,
> $$
>
> where each diagonal activation-derivative matrix $D_l(x)$ has norm $\leq 1$.
> Since weight decay penalizes Frobenius norms and
> $$
> |W_l|_F \ge |W_l|_2 ,
> $$
> it encourages small spectral norms and thus promotes local contraction [a,b].
>
> We included the full derivation in Appendix C of the paper for the weight decay case.
>
> Other links between contractive solutions and regularization are for example:
> * **Denoising Autoencoders.**
>   Training with Gaussian noise is known to enforce contraction around each training point
>   (Vincent et al., 2011; Alain & Bengio, 2012).
>
> * **Contractive Autoencoders.**
>   Directly penalize ($|J_f(x)|_F^2$), explicitly enforcing local contraction
>   (Rifai et al., ICML 2011).
>
> * **Other autoencoder variants.**
>   As summarized in Table 2, regularizers such as sparsity penalties, and VAE-style priors commonly bias the learned mapping toward contractive behavior.
>
> A formal proof for *all* regularization schemes are outside the scope of this work. Our point is that these training mechanisms tend to bias solutions toward contractive mappings, which we confirm empirically by consistently observing stable latent attractors and never encountering divergent trajectories in our experiments.
>
>
> _**[a]** Neyshabur et al., *Norm-based capacity control in neural networks*, COLT 2015._
>
> _**[b]** Bartlett et al., *Spectrally-normalized margin bounds for neural networks*, NeurIPS 2017._
>
> ---
>
> ### **Smoothness assumptions**
>
> We agree that smoothness of the latent distribution may not hold globally in all practical settings. However this can be explicitly promoted in large-scale models via (i) variational regularization, as in Stable Diffusion’s VAE backbone (used in the experiments in Section 4.1), where the KL term enforces smoothness and locally Gaussian latent structure (Kingma & Welling, 2013; Rombach et al., 2022); (ii) implicit inductive biases common at scale,weight decay (Krogh & Hertz, 1991; Loshchilov & Hutter, 2017), masking (He et al., 2022), residual connections (He et al., 2016), all of which constrain the spectral norm of the Jacobian.
>
> Empirically, our results (Fig. 2–6) show stable latent trajectories even in large pretrained models (Stable Diffusion and ViT-MAE), supporting that smoothness holds in practice within the support of the learned latent distribution.
>
> **We performed a lightweight empirical check of the smoothness of latent-trajectories** for the experiment in section 4.2. We measured directional changes along the update path using second-order finite differences (a standard curvature-based proxy in discretized trajectory analysis). Lower values indicate smaller changes in direction; we compare against the straight-line connecting the start and point of the trajectory (smoothest possible trajectory) and isotropic random walks with step scale matched to the velocities of our trajectories.
>
> | Trajectory Type             | Curvature Score (↓) |
> | --------------------------- | ------------------- |
> | Straight Line (baseline)    | ~0                  |
> | Random Walk (matched scale) | **2.00 ± 0.00**     |
> | Latent trajectories            | **0.85 ± 0.11**     |
>
> This provides an indicative sanity check that latent dynamics exhibit structured, smooth evolution rather than random fluctuation.

---

> ### Author Response · Authors · 2025-11-23
> **Rebuttal (2 of 2)**
>
> ### **Numerical validation of Proposition 3.2.**
>
>
> We validate proposition 3.2 in the context of the experiment performed in Figure 2 of the paper. We plotted the error decomposition of the MNIST models reported here as a Table for convenience. We observe:
> * The coverage error starts very high and decreases as an inverse function of regularization strength.
> * The prototype error starts low and increases til plateauing
> * The variance across samples of both error terms is much higher, as memorized points will have both low prototype error and coverage
>
>
>
> | metric | dim=2 | dim=4 | dim=8 | dim=16 | dim=32 | dim=64 | dim=128 | dim=256 | dim=512 |
> | :---: | :---: | :---: | :---: | :---: | :---: | :---: | :---: | :---: | :---: |
> | MSE | 0.37 ± 0.21 | 0.25 ± 0.17 | 0.13 ± 0.10 | 0.06 ± 0.04 | 0.03 ± 0.02 | 0.02 ± 0.01 | 0.01 ± 0.01 | 0.01 ± 0.01 | 0.01 ± 0.01 |
> | prototype error | 0.67 ± 0.31 | 1.34 ± 0.70 | 1.83 ± 0.80 | 1.92 ± 0.48 | 2.11 ± 0.53 | 1.64 ± 0.43 | 1.91 ± 0.34 | 2.07 ± 0.22 | 1.91 ± 0.29 |
> | coverage error | 9.82 ± 5.71 | 9.37 ± 4.98 | 8.16 ± 4.68 | 6.16 ± 1.92 | 4.00 ± 1.24 | 0.70 ± 0.22 | 0.14 ± 0.03 | 0.14 ± 0.02 | 0.05 ± 0.01 |
> | memorization | 0.93 ± 0.02 | 0.92 ± 0.03 | 0.85 ± 0.06 | 0.78 ± 0.07 | 0.63 ± 0.07 | 0.51 ± 0.11 | 0.37 ± 0.08 | 0.31 ± 0.03 | 0.37 ± 0.07
>
> This supports that memorizing models will have lower prototype error and narrow coverage, while generalizing models will cover better the latent space.
>
> ---
>
> ### **Iteration complexity under weaker assumptions.**
>
>
> We gently remark that the conditions in **Proposition A.10** already apply to **general differentiable networks**: the result only requires a **local contraction** ( $|J_f(z^*)|_\sigma < 1$) in a neighborhood of the fixed point. Under this assumption, the standard contraction argument yields
>
> $$
> |z_t - z*| \le L^t |z_0 - z*|,
> $$
>
>
> $$
> T \geq \frac{\log(\varepsilon/|z_0 - z^*|)}{\log L},
> $$
> which holds regardless of whether the network is linear, piecewise-linear, or fully nonlinear.
>
> In the **linear** and **piecewise-linear** cases, the local Lipschitz constant (the maximal singular value of the active linear map) can be computed **in closed form**, making the contraction factor and iteration bound exactly computable in each linear region. For **general nonlinear networks**, computing ($|J_f(z)|_\sigma$) is harder, but there is a substantial literature providing practical **upper bounds or estimates** of local and layerwise Lipschitz constants [a,b,c,d].
>
> _**[a]** Virmaux & Bottou. *Lipschitz Regularity of Deep Neural Networks: Analysis and Efficient Estimation.* NeurIPS 2018._
>
> _**[b]** Farnia et al . *Generalizable Adversarial Training via Spectral Normalization.* ICML 2018._
>
> _**[c]** Gouk et al. *Regularisation of Neural Networks by Enforcing Lipschitz Continuity.* ICLR 2021._
>
> _**[d]** Sedghi,et al *The Singular Values of Convolutional Layers.* NeurIPS 2019._

---

> > ### Comment · Reviewer_KKEL · 2025-11-27
> > **Reply**
> >
> > Thank you for the thorough and clear rebuttal, as well as the additional analyses. My understanding of the work has significantly improved, and I now find it to be a very good paper. I will therefore keep my original score (8: accept) unchanged.

---

> > > ### Author Response · Authors · 2025-11-28
> > >
> > > We thank the reviewer for following up and for noting that the rebuttal helped clarify questions and mase the contribution now coming across as a very good paper. We appreciate the constructive suggestions throughout the discussion.
> > >
> > > We remain available for any further questions for the reminder of the discussion period.

---

### Author Response · Authors · 2025-12-03

We sincerely appreciate the time and effort invested by the AC and the reviewers in assessing our submission. Given the unusual discussion circumstances, we provide a concise summary of how the paper evolved during the review process. Our paper was **already positively received before rebuttal**, and during the discussion **reviewers b56C and XuAT raised their scores to 8**, while **KKEL reaffirmed their initial 8**. The remaining reviewer, **LZxN**, did not participate in the discussion before the interruption, but had also provided a **positive initial assessment**. Feedback during rebuttal was **uniformly positive**.

We summarize below how we addressed all general concerns shared across reviewers; all additional reviewer-specific points were addressed as well in our rebuttal.

* **Extending the scope beyond autoencoders**
  *(raised by b56C, LZxN, XuAT - reviewers b56C, XuAT considered this addressed. Reviewer LZxN did not engage.)*
  We clarified that *Theorem 1* and *Proposition 3.1* apply to any **locally contractive self-map**, not only to autoencoders. To substantiate this, we added new experiments (Appendix B.8) on **self-supervised encoders (DINOv2, SigLIP2)** and **LLMs (Qwen3-0.6B, SmolLM2)**. In all cases, the induced maps exhibited **bounded, stable, and contractive latent trajectories**, showing preliminary that latent vector fields naturally arise well beyond AE architectures.

* **Expanding empirical validations of the theoretical claims**
  *(raised by KKEL and XuAT - all reviewers considered this addressed.)*
  We added convergence analyses of latent trajectories in **ViT-MAE** demonstrating contraction in large pretrained models, as well as an empirical validation of the **prototype–coverage decomposition** in Proposition 3.2, illustrating how the two error components relate to  memorization and generalization regimes.

* **Strengthening OOD baselines and ablations**
  *(raised by b56C and XuAT - all reviewers considered this addressed.)*
  We expanded the comparisons in Section 4.2 to include **Mahalanobis distance**, **reconstruction error**, along with sensitivity analyses on the choice of K in the KNN baseline. Across datasets, **trajectory-based attractor statistics** confirmed to be more informative for characterizing distribution shift, demonstrating that latent dynamics capture additional signal not available in static embeddings or reconstruction scores alone.

Reviewers noted that additions and clarifications in the rebuttal **substantially improved the clarity, rigor, and scope** of the work. With three reviewers converging to a **8**, the revised manuscript reflects the feedback gathered throughout the review process.

Sincerely,

Authors of submission 18083

---

### Meta-Review · Area_Chair_1rr9 · 2026-01-06

**Summary:**

This paper aims to interpret neural networks through the lens of dynamical systems. Specifically, the authors treat the networks (initially autoencoders, but later generalizing to other models) as operators over a vector space and proceed to compute properties of these operators. The authors show that this framework can predict generalization, memorization, and identify out-of-distribution data. The reviewers were all initially positive about this work. The concerns raised mostly had to do with questions of assumptions (e.g., is smoothness of the map defined by the network reasonable), justification of why properties arise (e.g., the contractive nature of the operators), impact of the theory beyond autoencoders, and clarifications on the details of the input prediction and other experimental details.

The authors did a thorough job addressing these concerns, and given that the primary concerns about theoretical justification and validity of assumptions were addressed, I view those as a the primary challenge to the paper. The smaller details were also addressed, to the satisfaction of 3/4 reviewers (one did not respond). I think that this remains a strong paper.

**Reviewer Concerns:**

The reviewers were fairly thorough in their response. Specifically they addressed:
1) the concern of impact outside of autoencoders through the inclusion of self-supervised encoders and next-token LLMs.
2) the concern about limited out-of-distribution assessment with new baseline comparisons.
3) the concern of the smoothness assumption by assessing curvature of different trajectories and discussing how stronger smoothness can be induced through regularization
4) the concern of where the contractiveness of the map comes from by adding more examples int he appendix to detail that implicit regularization promotes contractive maps.
5) various clarity issues through text edits.

While i think one or two details (e.g., the smoothness concern) could be explored a bit more, I don't think these are really weaknesses in the paper and the work was significantly strengthened with no outstanding concerns that I find detrimental to the work as presented.

**Reviewer Scores:**

The initial scores of this paper were 6,8,6,6. Of these, before the discussion was cut short, Reviewers KKEL and b56C both raised their scores from a 6 to an 8 while Reviewer XuAT responded to assert that their score stayed at 8. Reviewer LZxN did not respond, however given the strength of the responses and the increase across the board, I'd guess that there is at least a 50/50 chance that this reviewer would have also raised their score to an 8.

---

### Decision · Program_Chairs · 2026-01-26

Accept (Oral)